# An Analysis of Model Robustness across Concurrent Distribution Shifts

**Myeongho Jeon**[*]                                                                *myeongho.jeon@epfl.ch*
*École Polytechnique Fédérale de Lausanne*

**Suhwan Choi**[*]                                                                    *schoi828@snu.ac.kr*
*Seoul National University*
*CRABs.ai*

**Hyoje Lee**                                                                    *hyoje.lee@samsung.com*
*Samsung Research*

**Teresa Yeo**[†]                                                              *teresa.yeo@smart.mit.edu*
*Singapore-MIT Alliance for Research and Technology*

**Reviewed on OpenReview:** *https://openreview.net/forum?id=nxQtoHHcj9*

**Code:** *https://github.com/schoi828/robustness*

## Abstract

Machine learning models, meticulously optimized for source data, often fail to predict target data when faced with distribution shifts (DSs). Previous benchmarking studies, though extensive, have mainly focused on simple DSs. Recognizing that DSs often occur in more complex forms in real-world scenarios, we broadened our study to include multiple concurrent shifts, such as unseen domain shifts combined with spurious correlations. We evaluated 26 algorithms that range from simple heuristic augmentations to zero-shot inference using foundation models, across 168 source-target pairs from eight datasets. Our analysis of over 100K models reveals that (i) concurrent DSs typically worsen performance compared to a single shift, with certain exceptions, (ii) if a model improves generalization for one distribution shift, it tends to be effective for others, and (iii) heuristic data augmentations achieve the best overall performance on both synthetic and real-world datasets.

## 1 Introduction

Machine learning models deployed in real-world settings may face complex distribution shifts (DSs). These shifts can result in unreliable predictions. For instance, a self-driving car may be deployed in a place where there is a different driving etiquette, unfamiliar environment *and* weather condition. Thus, a systematic evaluation of such complex shifts is essential before deploying models in the real world.

Most research has focused on evaluating models under a single DS (**UniDS**) (Taori et al., 2020; Gulrajani & Lopez-Paz, 2020; Koh et al., 2021; Wiles et al., 2022; Miller et al., 2021; Wenzel et al., 2022). For instance, in the PACS evaluation benchmark (Li et al., 2017), the training data consists of photographs, but the model is tested on sketches. Similarly, in the training data of Biased FFHQ (Lee et al., 2021), gender is spuriously correlated to age, but the model is tested on data where gender is anti-correlated to age. In a real-world scenario, we can encounter these two shifts *concurrently*, i.e., one where age and gender are spuriously correlated *and* where there is also a shift in the style of the image.

---

[*]Equal contribution.
[†]Corresponding author.

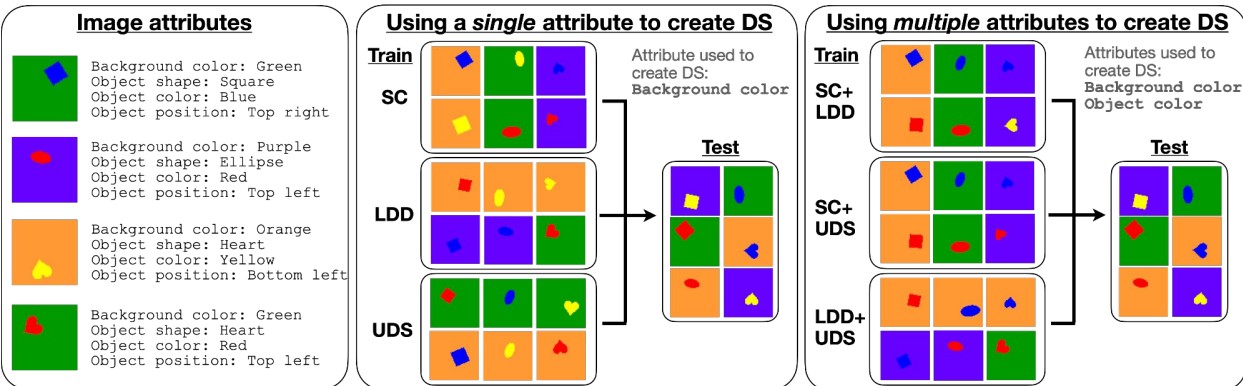

Figure 1: **Concurrent distribution shifts**. *Left*: We list some attributes of a few images from the **dSprites** dataset. In this dataset, the object shape is the label. *Center*: We show how a *single* attribute e.g., the background color, can be used to create different types of distribution shifts. Namely, spurious correlation (SC), where in this example, the background color is correlated with the object shape, low data drift (LDD), and unseen data shift (UDS). We assume that the test data consists of images where all attribute instances are equally likely to appear, i.e., each image is generated by randomly selecting each attribute instance with equal probability. *Right*: As the real world consists of more complex shifts, we also make use of *multiple* attributes to create combinations of distribution shifts. In the examples above, we use the background and object color to create combinations of distribution shifts. The first shift is created using the background attribute and the second shift, the shape attribute i.e., SC+UDS is created from a correlation between the background color and the shape (SC), ***and*** only using a subset of colors, for the shape attribute (UDS).

To address this, we introduce an evaluation framework that mirrors the complex DSs potentially found in real-world settings. We assume that the input is generated from a set of attributes, e.g., gender, age, image style, etc. Thus, we leverage existing multi-attribute datasets, such as CelebA (Karras et al., 2017), which includes 40 attribute labels such as gender, hair color, and smiling, to manipulate the type and number of DSs present in the dataset. We refer to this framework as **ConDS**. It can account for *the co-occurrence of different DSs across multiple attributes within paired source and target datasets.* An example of this protocol is illustrated in Figure 1. Consequently, evaluating existing methods against these shifts can provide insights unattainable with current benchmarks. Specifically, our proposed framework consists of the following components:

*Distribution Shifts.* We explore seven types of DSs, including spurious correlation (SC), low data drift (LDD), unseen data shift (UDS), and their combinations; that is, (SC, LDD), (SC, UDS), (LDD, UDS), and (SC, LDD, UDS). To offer a comprehensive analysis covering a wide range of cases, we select various attributes to induce DSs. This includes adjusting three attributes for each **UniDS** and creating six combinations from three attributes for each **ConDS**, resulting in 33 distinct cases per dataset.

*Algorithms.* We evaluate 26 different methods from 7 popular approaches: architectural strategies, data augmentations, de-biasing, worst-case generalization, single domain generalization, out-of-distribution generalization, and even zero-shot inference using vision-language foundation models.

*Experiments.* We evaluate these algorithms on all proposed combinations of DSs. This results in over 100K experiments across 168 dataset pairs sourced from 3 synthetic and 5 real-world benchmarks. Further, we examine aspects such as the effectiveness of pre-training and the influence of prompts on foundation models for image classification.

As a result, we have made some intriguing findings: (i) Concurrent shifts are generally more challenging than single shifts. However, in ConDS, when a particularly difficult DS is combined with relatively easier ones, the harder shift tends to dominate, limiting any further increase in overall difficulty. (ii) If a model improves generalization for one DS, it proves effective for others, even if it was originally designed to address a specific shift. (iii) Heuristic augmentation techniques outperform meticulously crafted generalization

methods overall. (iv) Although vision-language foundation models (e.g., CLIP, LLaVA) perform well on simple datasets, even in the presence of DSs, their performance significantly deteriorates on more complex, real-world datasets.

## 2 Related Work

In this section, we review existing benchmarks for distribution shifts (DSs) and frameworks for evaluating model generalization.

**Benchmarks.** Several types of benchmarks have been introduced to evaluate the generalization of models. One type of benchmark evaluates generalization on datasets collected from different sources, i.e., due to different domains (Li et al., 2017; Venkateswara et al., 2017; Peng et al., 2019; Koh et al., 2021) e.g., the train and test data are collected from different countries, or different time periods (Yao et al., 2022; Hendrycks et al., 2021). Another popular type applies transformations to the input (Hendrycks & Dietterich, 2019; Sagawa et al., 2019; Nam et al., 2020; Bahng et al., 2020; Jeon et al., 2022b). Such transformations range from analytical ones like rotation, corruptions like 'brightness' or 'contrast', to learned ones like adversarial attacks. Images with the same transformations form a domain, while each transformation can be an attribute, creating spurious correlations by matching labels to transformations. Other alternatives includes using engines like Blender or Unity to create simulators that can render different types of shifts (Leclerc et al., 2022; Sun et al., 2022). In contrast, Recht et al. (2019) collects new test sets for ImageNet and CIFAR-10 by replicating the data collection pipeline. They showed that without any explicit distribution shift, there is a drop in accuracy. Similarly, Barbu et al. (2019) collects a new dataset where objects have unusual poses or viewpoints. The benchmarks most similar to ours involve changing the original train-test split of the dataset to induce different types of DSs (Kim et al., 2019; Koh et al., 2021; Jeon et al., 2022a; Atanov et al., 2022; Jeon et al., 2022b). In contrast to these methods, our paper focuses on creating controllable concurrent shifts by using attribute annotations in existing datasets.

**Large-scale Generalization Analysis.** Although methods that enhance robustness against DSs have been extensively researched, significant variations across application domains mean that a method that excels in one dataset might not perform equally well in another. Consequently, recent efforts have been dedicated to comprehensively and fairly evaluate generalization methods. Gulrajani & Lopez-Paz (2020) demonstrated that empirical risk minimization, when meticulously implemented and finely tuned, excels over domain generalization methods in robustness. Taori et al. (2020) found that generalizability to synthetic shifts does not guarantee robustness to natural shifts. Koh et al. (2021) introduced **Wilds**, a curated benchmark consisting of 10 datasets that encapsulate a diverse range of DSs encountered in real-world settings. Their findings indicate that current methods for generalization are inadequately equipped to address real-world DSs. Wiles et al. (2022) reported that both pre-training and augmentations significantly boost performance across many scenarios, although the most effective methods vary with different datasets and DSs. Additionally, Miller et al. (2021) and Wenzel et al. (2022) observed a positive correlation between out-of-distribution and in-distribution performance. However, these studies predominantly concentrate on single DSs, which do not fully capture real-world scenarios. Ye et al. (2022) categorized existing benchmarks based on the extent of spurious correlations and the degree of domain shift. Their findings reveal that, for the most part, Out-of-Distribution generalization algorithms remain susceptible to spurious correlations.

## 3 Problem Statement

In this section, we introduce the concept of distribution shift (DS) and its two categories: unique distribution shift (**UniDS**) and concurrent distribution shift (**ConDS**). To operationalize the concept of **ConDS**, annotations for multiple attributes are required. For clarity, we begin by introducing one of our experimental datasets, dSprites, in Figure 2.

### 3.1 Distribution Shift

Consider an instance set $X = \{\mathbf{x}_i | \mathbf{x}_i \in \mathcal{X}, i = 1, \ldots, N\}$ for a classification problem, where $\mathcal{X}$ denotes the input space. Each instance $\mathbf{x}$ can be represented by a finite set of attributes $Y(\mathbf{x}) = \{y^k | y^k \in \mathcal{Y}, k = 1, \ldots, K\}$ where $\mathcal{Y}$ denotes the attribute space and $K$ varies with $\mathbf{x}$. Within this framework, one attribute $y^l$ from $Y(\mathbf{x})$ can be designated as the label. Let $p$, $p_S$, and $p_T$ be the true, source, and target distributions, respectively. We denote the source and target datasets by $\mathcal{D}_S \sim p_S$ and $\mathcal{D}_T \sim p_T$, respectively, where each dataset consists of samples from its corresponding distribution. A machine learning model $\hat{f}$ is designed to minimize the empirical risk,

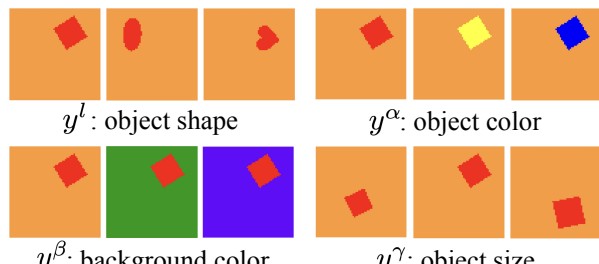

$y^l$: object shape      $y^\alpha$: object color

$y^\beta$: background color      $y^\gamma$: object size

Figure 2: **dSprites samples.** Even in simple synthetic data, multiple attributes can potentially lead to various DSs. Visualizations for other datasets are included in Figure 5 in the Section A.1.

$$\hat{f} := \arg\min_f \frac{1}{N} \sum_{i=1}^{N} \mathcal{L}(y_i^l, f(\mathbf{x}_i)), \tag{1}$$

where $(\mathbf{x}_i, y_i^l) \in \mathcal{D}_S$ for $i = 1, ..., N$. The ideal objective is to find the model $f^*$ that maximizes its performance on true distribution $p(\mathbf{x}, y)$; that is,

$$f^* := \arg\min_{\hat{f}} \mathbb{E}_{p(\mathbf{x}, y^l)} \left[ \mathcal{L}(y^l, \hat{f}(\mathbf{x})) \right]. \tag{2}$$

However, a significant shift in distribution from source to target data, denoted by $\mathcal{D}_S \not\approx \mathcal{D}_T$, can cause a substantial decrease in performance during inference. Assuming that the target distribution $p_T$ aligns with $p$, we define DSs in the following subsections.

### 3.2 Unique Distribution Shifts

We revisit spurious correlation (SC), low data drift (LDD), and unseen data shift (UDS) as delineated in the experimental framework by Wiles et al. (2022), grouping these under the category of unique distribution shifts (**UniDS**), namely **UniDS** := {SC, LDD, UDS}. To create the **UniDS**, we select one attribute from the candidates ($y^\alpha$, $y^\beta$, $y^\gamma$) and then sample the images for each DS accordingly. All other attributes display a uniform distribution for **UniDS**.

**Test distribution $p_T$.** All the attributes $y^k \in Y(\mathbf{x})$ are uniformly distributed, ensuring that each attribute is represented and independent from the others. For example, for $p_T \approx p$, there is an equal number of samples across 81 combinations, ranging from (square, red, orange, small) to (heart, blue, purple, big).

**Spurious correlation.** Under $p_S$, the label $y^l$ and a specific attribute $y^\alpha$ exhibit correlation, which does not hold under $p_T$. For example, there are only (square, red), (ellipse, yellow), and (heart, blue) samples in $\mathcal{D}_S$, exhibiting a correlation between 'shape' and 'color'.

**Low data drift.** Attribute values show an uneven distribution under $p_S$ but are more evenly distributed under $p_T$. Generalization for LDD is also referred to as worst-case generalization (Sagawa et al., 2019; Seo et al., 2022). For example, there are significantly fewer red samples compared to the numerous yellow and blue samples. Even among the red samples, the distribution of shapes is uneven.

**Unseen data shift.** Some attribute values that are not present under $p_S$ appear under $p_T$. E.g., There are no blue images in $\mathcal{D}_S$, indicating that blue images in $\mathcal{D}_T$ are new to the model.

### 3.3 Concurrent Distribution Shifts

Previous benchmarking studies have predominantly focused on **UniDS**. However, in real-world contexts, a dataset could contain concurrent DSs. Given that an instance $\mathbf{x}$ encompasses numerous attributes, DSs

can exhibit more complex structures, with each attribute governed by distinct **UniDS**. For instance, in the real-world dataset iWildCam, UDS are observed due to discrepancies in the camera traps used across source and target data. Concurrently, LDD emerges from variable image counts across animal classes. Therefore, to effectively assess the efficacy of methods in real-world applications, it is crucial to conduct a comprehensive evaluation that addresses these scenarios.

To reflect the complexities observed in real-world datasets, we introduce the concept of "concurrent distribution shift (**ConDS**)". This novel framework extends conventional **UniDS** by modeling the diverse shifts associated with different attributes of a dataset. In this study, we focus on combining various shifts attributed to different attributes, rather than evaluating a single type of shift across multiple attributes—a topic extensively covered in previous literature.

> In our formal definition, a **ConDS** is conceptualized as the ensemble of all possible combinations of two or more distinct **UniDS** elements,
>
> $$\mathbf{ConDS} := \{S \subseteq \mathbf{UniDS} \mid |S| \geq 2\}, \tag{3}$$
>
> where each subset $S$ represents a unique configuration of shifts, creating a richer and more representative model of the underlying complexities in real-world data. Each pair $(y^l, y^k)$ within this framework, where $k$ belongs to a predefined set $\{\alpha, \beta, \gamma\}$, is governed by specific **UniDS**.

For instance, using the attributes shown in Figure 2, (SC, UDS) can be established by sampling combinations such as (square, red), (ellipse, yellow), and (heart, blue), while the background color varies between (orange, green) and the object size maintains a uniform distribution. The concept of **ConDS** and its importance were also introduced in Koh et al. (2021). However, their discussion was limited to LDD and UDS. We broaden this to include SC, LDD, and UDS, aiming for a thorough understanding of their interconnections. Additionally, we investigate a broader array of algorithms to assess how existing methods perform on **ConDS**.

## 4 Generalization to Distribution Shifts

As discussed in Section 3.1, it is assumed that during training, we only have access to source data, while the target distribution remains unknown (Vapnik, 1991; Jeon et al., 2023). While some algorithms are specifically designed for scenarios where partial knowledge of the target distribution is available (Adeli et al., 2021; Alvi et al., 2018; Kim et al., 2019; Geirhos et al., 2018; Bahng et al., 2020; Ganin et al., 2016), the setup where no target information is known generally poses a broader challenge for model generalization. Consequently, our analysis focuses on this more general setup. We evaluate 26 algorithms suitable for this scenario, spanning a wide spectrum of methods, as shown in Table 1.

Table 1: **Generalization algorithms evaluated.** We outline the methods evaluated and the distribution shifts they are designed to address.

| Generalization Algorithms | | SC | LDD | UDS |
|---|---|:---:|:---:|:---:|
| *Architecture* | ResNet18, ResNet50, ResNet101 (He et al., 2016), ViT (Dosovitskiy et al., 2020), MLP (Vapnik, 1991). | | | |
| *Heuristic augmentation* | Imagenet (He et al., 2016), AugMix (Hendrycks et al., 2019), RandAug (Cubuk et al., 2020), AutoAug (Cubuk et al., 2019). | | | ✓ |
| *De-biasing* | UBNet (Jeon et al., 2022a), PnD (Li et al., 2023), OccamNets (Shrestha et al., 2022). | ✓ | | |
| *Worst-case generalization* | GroupDRO (Sagawa et al., 2019), BPA (Seo et al., 2022). | | ✓ | |
| *Single domain generalization* | ADA (Volpi et al., 2018), ME-ADA (Zhao et al., 2020), SagNet (Nam et al., 2021), L2D (Wang et al., 2021). | | | ✓ |
| *Out-of-distribution generalization* | IRM (Arjovsky et al., 2019), CausIRL (Chevalley et al., 2022). | ✓ | ✓ | ✓ |
| *Zero-shot inference with foundation model* | CLIP (Radford et al., 2021), InstructBLIP (Dai et al., 2024), LLaVA-1.5 (Liu et al., 2023), Phi-3.5-vision (Abdin et al., 2024), GPT-4o-mini, GPT-4o (OpenAI, 2024). | | | |

**Augmentation for generalization.** Although some augmentation techniques are not specifically designed for robustness, they are commonly used to enhance generalization against the unseen domain and are well known for their effectiveness (Yan et al., 2020; Wiles et al., 2022; Cugu et al., 2022; Zheng et al., 2024). The operative idea is that augmentation expands the input space, thereby increasing the likelihood that the model will recognize new test samples.

**Generalization algorithms.** *De-biasing*: UBNet utilizes feature maps from lower to higher layers to enable the model to access a broader feature space. PnD removes spurious correlations by detecting them using GCE (Zhang & Sabuncu, 2018). OccamNets employs architectural inductive biases, demonstrating their effectiveness in mitigating spurious correlations. *Worst-case generalization*: GroupDRO and BPA modify the weight of gradient flow to ensure balance across different groups. *Single domain generalization*: ADA and ME-ADA generate additional inputs by incorporating adversarial noise. SagNet is designed to be agnostic to styles, emphasizing content instead. L2D utilizes learned augmentations with AdaIN from StyleGAN (Karras et al., 2019). *Out-of-distribution generalization*: IRM and CausIR eliminate variant features by regularizing discrepancies among samples from the source data.

**Foundation model for image classification.** Foundation models acquire robust representations from comprehensive datasets, facilitating their application in downstream tasks such as image classification. The mechanisms for aligning inputs to outputs ($\mathcal{X} \to \mathcal{Y}$) during zero-shot inference differ among models. CLIP utilizes the similarity between image embeddings and textual label embeddings to calculate confidence scores. Conversely, InstructBLIP, LLaVA-1.5, Phi-3.5-vision, and GPT-4o employ label-eliciting prompts as queries within the framework of visual question answering (Chen et al., 2022). Additional details about the prompts used for image classification can be found in Table 12 of the Section B.5.

## 5 Experiments

In this section, we begin by presenting the datasets we evaluated on and the experimental setup. Next, we evaluated 26 distinct algorithms across 168 (source, target) pairs spanning six datasets, addressing both **UniDS** and **ConDS**. The aggregate results are illustrated in Figure 3, while the comprehensive results are detailed in Section B.1. We further examine how the challenges vary among different DSs (Figure 4) and evaluate the effectiveness of pre-training in enhancing robustness (Table 3). Further analysis investigates how zero-shot inference performance depends on distribution shifts (DSs), as shown in Table 5. Finally, we analyze the outcomes, summarizing them into eight key takeaways.

### 5.1 Datasets

**Controlled datasets:** We assess algorithms using five evaluation datasets: **dSprites**, **Shapes3D**, **Small-Norb**, **CelebA**, and **DeepFashion**. From these, we select four attributes: one is designated as the label ($y^l$), and the other three as attributes ($y^\alpha, y^\beta, y^\gamma$) to create DSs. Table 2 lists the attribute instances for $y^l$ and $\{y^\alpha, y^\beta, y^\gamma\}$ for the different controlled datasets. We divide the source data into training and validation sets, both sharing the same distribution, with the validation set used for hyperparameter tuning.

**Uncontrolled real-world datasets:** We use **iWildCam**, **fMoW**, and **Camelyon17** for evaluation. **iWildCam** data in the Wilds benchmark (Koh et al., 2021) exhibits LDD over the animal distributions, and UDS occurs across camera trap locations. Similarly, the **fMoW** dataset (Koh et al., 2021) exhibits UDS and LDD across time and regions in satellite images. **Camelyon17** is a tumor detection dataset with various unexpected DSs resulting from different hospitals. We further discuss additional insights from the categorization of synthetic (**Dsprites**, **Shapes3D**, **Smallnorb**) and real-world datasets (**CelebA**, **DeepFashion**, **iWildCam**, **fMoW**, **Camelyon17**) in Section A.3.

### 5.2 Experimental Setup and Results

**Controlled datasets:** To comprehensively understand the generalizability of the algorithms on diverse DSs, we evaluate with three different seeds for five datasets changing attributes $\{y^\alpha, y^\beta, y^\gamma\}$. Specifically, with labels fixed to $y^l$, we manipulate attributes to simulate different DSs. For instance, we select one attribute

Table 2: **Attributes for controlled datasets.** Each dataset's attributes and their possible values. Image samples from each dataset are displayed in Figure 5 in Section A.1.

| Dataset | | Attribute | Values |
|---|---|---|---|
| dSprites | $y^l$ | object shape | {square, ellipse, heart} |
| | $y^\alpha$ | object color | {red, yellow, blue} |
| | $y^\beta$ | background color | {orange, green, purple} |
| | $y^\gamma$ | object size | {small, middle, big} |
| Shapes3D | $y^l$ | object shape | {cube, cylinder, sphere, capsule} |
| | $y^\alpha$ | object color | {red, orange, yellow, green} |
| | $y^\beta$ | background color | {red, orange, yellow, green} |
| | $y^\gamma$ | object size | {tiny, small, middle, big} |
| SmallNorb | $y^l$ | object | {animal, human, car, truck, airplane} |
| | $y^\alpha$ | azimuth | {0, 80, 160, 240, 320} |
| | $y^\beta$ | lighting | {0, 1, 2, 3, 4} |
| | $y^\gamma$ | elevation | {30, 40, 50, 60, 70} |
| CelebA | $y^l$ | gender | {male, female} |
| | $y^\alpha$ | hair color | {black, others} |
| | $y^\beta$ | smiling | {smiling, no smiling} |
| | $y^\gamma$ | hair style | {straight, others} |
| DeepFashion | $y^l$ | dress | {skirt, others} |
| | $y^\alpha$ | pattern | {floral, solid} |
| | $y^\beta$ | sleeve | {long sleeve, sleeveless} |
| | $y^\gamma$ | fabric | {chiffon, cotton} |

from $\{y^\alpha, y^\beta, y^\gamma\}$ to create **UniDS** (3 settings). For **ConDS**, we use six combinations (either $_3C_2$ or $_3C_3$ settings) to establish designated DSs, resulting in a total of 165 (source, target) pairs. For example, for the (SC, LDD) setting, we choose one attribute to define SC and another from the remaining two to create LDD, resulting in 6 (source, target) pairs. The detailed description for this is provided in Section A.1. For all SC,

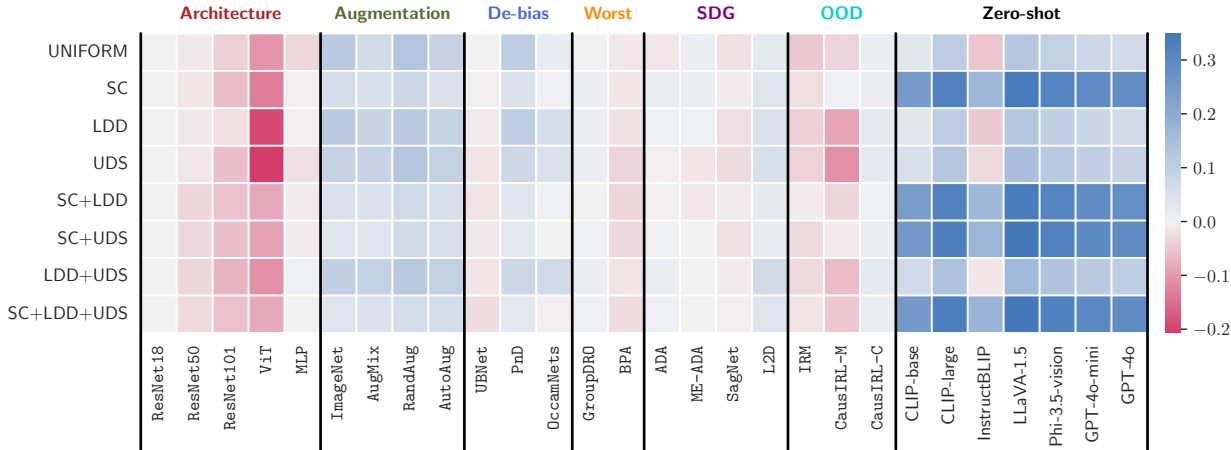

Figure 3: **Aggregate result on controlled datasets**. We plot the change in accuracy compared to the base model, ResNet18, averaged across all seeds and controlled datasets with varying attributes. Blue indicates improved performance, while red indicates a decline. Each row is independent of the others. The models used for *zero-shot inference* were only used for evaluation, thus, they have the same absolute performance for each row. However, as we show their accuracies relative to the ResNet18, the relative performance for each model is not the same for each row. We fine-tune all the algorithms and report their optimal results. Augmentation methods and zero-shot models perform well under the different types of shifts. We provide a breakdown of the accuracy for each algorithm and dataset in the Section B.1.

Table 3: **Training from scratch *vs* Pre-training.** All the values reported for controlled datasets represent the average accuracy across all DSs and controlled datasets. We compare the performance of models trained from scratch with those pre-trained, bolding the better result between the two training strategies. The top average result in each column except *zero-shot inference* is indicated with a gray box, while the second-best is underlined. We used the original models for *zero-shot inference* for evaluations, thus, they are neither trained from scratch nor fine-tuned. **Trends on controlled datasets *vs* Real-world datasets.** We also show the results from evaluating on **iWildCam**, **fMoW**, and **Camelyon17**, as they are challenging real-world datasets with **ConDS**. The experimental results from our controlled datasets align with those from the real-world datasets, demonstrating that even on real-world datasets, *augmentation and pre-training significantly enhance robustness.*

| | | Controlled Dataset | | Real-world Dataset | |
| --- | --- | --- | --- | --- | --- |
| | | *Scratch* | *Pre-training* | *Scratch* | *Pre-training* |
| ***Architecture*** | ResNet18 | 62.43(±0.84) | **81.53(±0.67)** | 53.91(±2.19) | **61.64(±1.71)** |
| | ResNet50 | 60.13(±0.85) | **82.64(±0.62)** | 52.58(±1.86) | **67.76(±1.46)** |
| | ResNet101 | 56.98(±0.88) | **79.96(±0.69)** | 47.01(±2.09) | **67.52(±1.41)** |
| | ViT | 49.55(±0.60) | **78.53(±0.62)** | 50.55(±2.11) | **70.94(±1.55)** |
| | ***avg.*** | *57.27(±2.81)* | *80.67(±0.90)* | *51.01(±1.50)* | *66.97(±1.94)* |
| ***Augmentation*** | *ImageNet* | 69.25(±0.88) | **80.55(±0.64)** | 48.82(±1.80) | **66.48(±1.60)** |
| | AugMix | 68.99(±0.83) | **83.15(±0.56)** | 51.99(±1.74) | **68.09(±1.53)** |
| | RandomAug | 71.47(±0.86) | **85.28(±0.53)** | 51.94(±1.84) | **68.29(±1.62)** |
| | AutoAug | 69.65(±0.85) | **85.85(±0.53)** | 52.15(±1.63) | **70.29(±1.59)** |
| | ***avg.*** | *69.84(±0.56)* | *83.71(±1.20)* | *51.22(±0.80)* | *68.29(±0.78)* |
| ***De-biasing*** | UBNet | 61.08(±0.86) | **78.26(±0.67)** | 41.01(±2.32) | **59.03(±1.77)** |
| | PnD | 68.15(±0.87) | **77.67(±0.79)** | 50.02(±2.49) | **60.34(±1.55)** |
| | OccamNets | 64.98(±0.71) | **77.59(±0.68)** | 52.57(±1.80) | **65.20(±1.39)** |
| | ***avg.*** | *64.74(±2.04)* | *77.84(±0.21)* | *47.87(±3.51)* | *61.52(±1.88)* |
| ***Worst-case generalization*** | GroupDRO | 63.26(±0.84) | **81.23(±0.67)** | 48.17(±1.87) | **64.40(±1.26)** |
| | BPA | 59.99(±0.81) | **75.69(±0.74)** | 40.41(±3.93) | **62.15(±1.69)** |
| | ***avg.*** | *61.63(±1.63)* | *78.46(±2.77)* | *44.29(±3.88)* | *63.28(±1.13)* |
| ***Single domain generalization*** | ADA | 62.98(±0.82) | **80.20(±0.68)** | 52.88(±2.02) | **69.71(±1.51)** |
| | ME-ADA | 62.37(±0.82) | **78.33(±0.69)** | 53.79(±2.04) | **66.41(±1.20)** |
| | SagNet | 61.09(±0.86) | **81.56(±0.67)** | 54.27(±1.93) | **66.75(±1.27)** |
| | L2D | 66.42(±0.83) | **82.63(±0.60)** | 52.67(±2.47) | **68.20(±1.95)** |
| | ***avg.*** | *63.22(±1.14)* | *80.68(±0.93)* | *53.40(±0.38)* | *67.77(±0.75)* |
| ***Out-of-distribution generalization*** | IRM | 59.83(±0.83) | **80.27(±0.68)** | 39.47(±2.42) | **57.90(±2.14)** |
| | CausIRL-M | 57.55(±0.98) | **80.81(±0.75)** | 48.90(±1.97) | **65.87(±1.21)** |
| | CausIRL-C | 64.17(±0.85) | **82.02(±0.68)** | 54.38(±2.07) | **68.06(±1.55)** |
| | ***avg.*** | *60.52(±1.94)* | *81.03(±0.52)* | *47.58(±4.35)* | *63.94(±3.09)* |
| ***Zero-shot inference*** | CLIP-base | 78.92 | | 25.76 | |
| | CLIP-large | 86.22 | | 29.06 | |
| | InstructBLIP | 70.95 | | 29.35 | |
| | LLaVA-1.5 | 88.24 | | 24.02 | |
| | Phi-3.5-vision | 85.63 | | 29.17 | |
| | GPT-4o mini | 83.62 | | 42.77 | |
| | GPT-4o | 82.57 | | 45.16 | |
| | ***avg.*** | *82.31* | | *32.18* | |

we include 1% of counterexamples, similar to the setup established in the research on SC robustness (Jeon et al., 2022a; Nam et al., 2020). We set ResNet18 as the backbone for all algorithms. More details about hyperparameters are provided in Section A.6. **Real-world datasets:** We adhered to the use of the train, validation, and test sets for **iWildCam**, **fMoW**, and **Camelyon17**. Real-world datasets do not exhibit a clear distribution shift like controlled datasets, but they inherently contain various naturally occurring distribution shifts that may go unnoticed.

**Results.** Figure 3 shows the aggregate result of algorithms across DSs for the controlled datasets. Figure 8 and Table 3 present the outcomes when methods are trained using ImageNet pre-trained weights. In Table 3, some algorithms, particularly augmentation techniques, surpass large models in scenarios without SC, an advantage not observed when training from scratch (see Figure 3). Table 3 indicates that pre-training

enhances performance for all the algorithms and DSs. In our experiments, while foundation models perform well on controlled datasets, their effectiveness is often limited when applied to real-world datasets.

**Comparison of distribution shifts.** In Figure 4, we evaluate the challenges associated with varying numbers of DSs. To ensure a consistent comparison across these shifts, we standardize the sizes of the datasets. However, due to limitations in data availability that prevent standardization while adhering to specific DSs, our analysis primarily relies on the **Dsprites** and **CelebA**. As illustrated in Figure 4, although the overall performance of algorithms tends to degrade as the number of DSs increases, SC remains inherently challenging and shows almost no performance decline when combined with other DSs. Additionally, SC is significantly more challenging than other distribution shifts, even when considering multiple shifts combined, such as LDD+UDS.

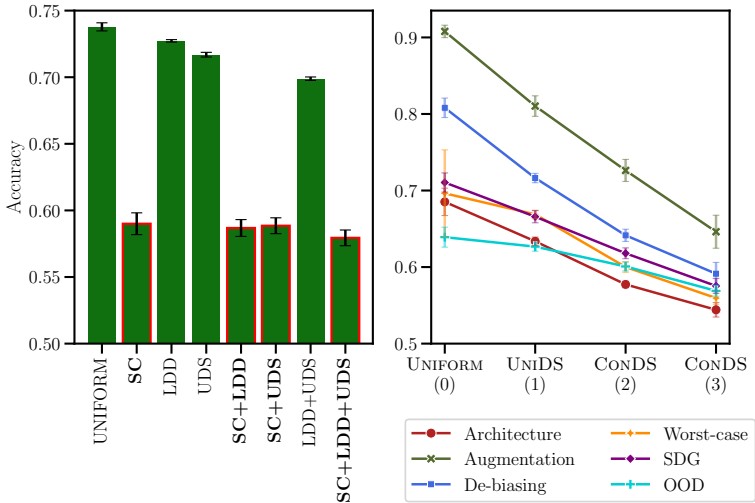

Figure 4: **Analysis of model robustness on distribution shifts.** *Left:* Average performance of all generalization methods under different combinations of DSs. *Right:* Comparing the different generalization methods under an increasing number of DSs.

Table 4: **An example of zero-shot inference prompts.** The first two rows show the 'General' prompts and the next three rows the 'Tailored' prompts that we used with LLaVA-1.5. For a more comprehensive list of these prompts, please see Table 12 in Section B.5.

| Type | Prompt |
|---|---|
| General$_1$ | *USER: <image>*
*Classify the image into label$_1$, . . . , or label$_C$. Please provide only the name of the label.* |
| General$_2$ | *USER: <image>*
*Choose a label that best describes the image. Here is the list of labels to choose from: label$_1$, . . . , label$_C$.*
*Please provide only the name of the label.* |
| **dSprites**
**Shapes3D**
**SmallNorb** | *USER: <image>*
*Classify the object in the image into label$_1$, ..., or label$_C$. Please provide only the name of the label.* |
| **CelebA** | *USER: <image>*
*Classify the person in the image into label$_1$ or label$_2$. Please provide only the name of the label.* |
| **DeepFashion** | *USER: <image>*
*Is a person wearing a dress or not? Please answer in yes or no.* |

**Further analysis on prompts of foundation model.** We investigate various prompts to assess the effectiveness of foundation models for robust image classification. We assess the accuracy of the models in

Table 5: **Comparison of the performance of visual language models** across different types of prompts. We report the best results obtained from this ablation study in Figure 3, 8 and Table 3. Note how sensitive the results are to the prompts used. There can be a drop in accuracy of 37% with a suboptimal prompt.

| Model | Prompt | dSprites | Shapes3D | SmallNorb | CelebA | DeepFashion | iWildCam | fMoW | Camelyon17 |
|---|---|---|---|---|---|---|---|---|---|
| CLIP-base | General | 75.19 | 71.41 | 72.02 | 98.5 | 77.50 | 13.97 | 13.30 | 50.01 |
| CLIP-large | General | 90.86 | 85.39 | 86.00 | 97.62 | 71.25 | 13.70 | 23.46 | 50.01 |
| InstructBLIP | General$_1$ | 50.25 | 50.08 | 26.59 | **98.63** | 61.25 | 1.32 | 11.61 | 50.57 |
| | General$_2$ | 53.46 | 53.05 | **36.22** | 70.50 | 47.50 | **1.86** | **18.17** | **68.03** |
| | Tailored | **58.89** | **74.77** | 25.41 | 78.25 | **86.25** | 1.10 | 9.67 | 50.00 |
| LLaVA-1.5 | General$_1$ | 49.01 | 52.89 | **90.05** | 98.50 | 70.00 | **4.64** | 15.04 | 50.57 |
| | General$_2$ | 72.72 | **74.61** | 87.81 | **98.88** | **91.25** | 1.60 | **16.32** | **51.09** |
| | Tailored | **86.42** | 71.02 | 88.16 | **98.88** | 88.75 | 1.85 | 14.24 | 50.00 |
| Phi-3.5-vision | General$_1$ | 88.27 | **72.73** | 83.20 | 98.62 | **82.50** | 5.88 | 9.67 | 64.86 |
| | General$_2$ | 88.27 | 72.50 | **85.41** | 98.75 | 78.75 | 4.06 | **10.88** | 57.09 |
| | Tailored | **88.77** | 72.58 | 83.30 | 98.62 | 53.75 | **9.91** | 9.17 | **66.71** |
| GPT-4o mini | General$_1$ | **96.79** | 94.77 | 81.12 | **93.87** | **50.0** | **43.43** | 18.87 | 64.81 |
| | General$_2$ | 92.84 | 94.77 | 78.37 | **93.87** | **50.0** | 43.36 | 19.28 | 64.87 |
| | Tailored | 92.84 | 94.77 | **82.69** | **93.87** | **50.0** | 41.88 | **19.89** | **64.98** |
| GPT-4o | General$_1$ | **92.84** | 94.77 | **80.22** | 82.87 | 50.00 | 51.64 | 24.98 | 58.02 |
| | General$_2$ | **92.84** | 94.77 | 78.37 | **93.75** | **51.25** | **51.80** | 24.89 | 58.03 |
| | Tailored | **92.84** | 94.77 | 78.85 | 89.62 | 50.0 | 51.21 | **25.58** | **58.10** |

Table 5. For CLIP, we use widely adopted prompts, such as "a photo of a *label*" (Matsuura et al.). Given that vision-language models designed for visual question answering may depend on the query's context, the choice of prompts is crucial (Sahoo et al., 2024). We categorize prompts that are applicable to any dataset as 'General', and those specifically designed for each dataset as 'Tailored'. We provide several examples for the prompt in Table 4. In Matsuura et al. and Islam et al. (2023), diverse prompts are employed as queries for vision-language models, specifically tailored to the dataset and the labels targeted for classification. Our analysis builds on this approach.

## 5.3 Takeaways

**Takeaway 1 – ConDS is, *on average*, more challenging than UniDS.** While previous studies on model generalization have primarily focused on **UniDS**, we observe that most algorithms exhibit poorer performance in **ConDS**. Moreover, the more numerous the DSs, the greater the challenge they pose, as illustrated in Figure 4 (right).

**Takeaway 2 – SC is the most challenging DS, followed by UDS and LDD.** Figure 4 (left) breaks down the performance of generalization methods according to DS. We see that the presence of SC tends to dominate over other DSs in **ConDS**. Although **ConDS** presents more challenges than **UniDS** for LDD and UDS, there is almost no performance drop when moving from SC to SC+LDD or SC+UDS. Furthemore, for the DSs that include SC, the performance of most methods is inferior to that of foundation models even when it is pre-trained, as shown in Figure 8.

**Takeaway 3 – Generalization tends to be consistent across DSs.** If a method improves generalization for one DS, it tends to be effective for others. Namely, although models such as *de-biasing*, *worst-case generalization*, and *domain generalization* are designed to address a specific DS (as detailed in Section 4), their applicability is not confined to that particular shift.

### 5.3.1 Takeaways Common to both UniDS and ConDS

The following takeaways with **UniDS** have been previously identified by Sahoo et al. (2024); Wiles et al. (2021). We observe that these phenomena persist even in the **ConDS** scenario.

**Takeaway 4 – Heuristic augmentations and pre-training are highly effective.** As depicted in Figure 3, heuristic augmentations overall improve the model's robustness across all DSs. This remains consistent when pre-trained weights are utilized except for ImageNet augmentation (Figure 8 in the supplementary). All the augmentation methods even increase the performance of ResNet18 when used on Uniform, where there is no DS, establishing them as a more effective tool. Algorithms demonstrate improved performance with ImageNet pre-trained weights, as shown in Table 3. On LDD, UDS, and LDD+UDS, most methods even outperform foundation models, as depicted in Figure 8.

**Takeaway 5 – Generalization algorithms provide a limited performance improvement.** While some methods (L2D, PnD, OccamNets) perform well, most methods exhibit limitations. Existing generalization methods, though competitive in their original experimental setups and benchmarks, demonstrate reduced effectiveness in our more standardized and fair setup.

**Takeaway 6 – Foundation pre-trained models are effective, but their performance can vary.** Although foundation models demonstrate impressive performance on controlled datasets, they show limited performance compared to other baselines with ImageNet-pretrained weights, as shown in Figure 8. Notably, they exhibit the lowest accuracy on real-world datasets (Table 3). This observation underscores that the performance in image classification significantly depends on the datasets used to train these foundation models, highlighting potential limitations in their applicability. E.g., for specialized datasets like **Camelyon17**, large models with zero-shot inference can barely make accurate predictions (Table 12). Furthermore, we find that their performances are significantly influenced by the prompt we input (Please refer Table 5 and Table 12 in the Section B.5 for more details).

## 6 Concluding Remark

*Contribution.* In this paper, we introduce a novel evaluation framework to understand the robustness of models against various distribution shifts, including **UniDS** and **ConDS**. Using this protocol, we can create distribution shifts in any multi-attribute-annotated dataset, allowing for a more comprehensive understanding of robustness. Through our extensive evaluation involving 100K experiments, we find that **ConDS** present greater challenges compared to conventional **UniDS**, with spurious correlations being more problematic than low data drift and unseen domain shifts. Our results indicate that heuristic augmentations and pre-training are effective tools for enhancing generalization, while more complex models offer limited benefits. Additionally, while large models are viable for image classification, their performance is effective only in specific scenarios and requires careful application.

*Limitation and future work.* While we believe that our work makes a promising step towards understanding how models behave under complex scenarios, there is still a lot more that can be done in this direction, we briefly discuss some of them. Our evaluation framework allows us to create controlled distribution shifts and assumes a uniform test distribution. Our framework also uses annotated, thus, interpretable attributes to create shifts. Using learned attributes would also be an interesting future direction. Furthermore, our study covers a limited range of attributes, particularly in controlled real-world datasets such as **CelebA** and **DeepFashion**, due to insufficient samples for other attributes. Future research could explore the use of advanced controllable generative models (e.g., diffusion models) to address this limitation and cover a broader range of conditions.

**Acknowledgments.** Myeongho Jeon, jointly affiliated with EPFL and Seoul National University, was supported by the National Research Foundation of Korea (NRF) grant funded by the Korea government (MSIT) [RS-2024-00337693]. Suhwan Choi was supported by the Ministry of Culture, Sports and Tourism & Korea Creative Content Agency [RS-2024-00399433], and Artificial intelligence industrial convergence cluster development project funded by the Ministry of Science and ICT (MSIT, Korea) & Gwangju Metropolitan City. We also extend our sincere thanks to CRABs.ai for their generous financial support and the provision of GPU resources.

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

# A  Detailed Experimental Setup

## A.1  Setup for Controlled Distribution Shifts

We utilize multi-attribute datasets such as **dSprites**, **Shapes3D**, **SmallNorb**, **CelebA**, and **DeepFashion** to develop **ConDS**. These datasets enable us to sample images annotated with various attributes, showcasing a range of DSs from SC to (SC, LDD, UDS). Additionally, we configure different attributes for each type of DS. For instance, in SC, pairs like $(y^\alpha, y^l)$ and $(y^l, y^\beta)$ may exhibit spurious correlations. By covering all possible attribute combinations (with the attributes depicted in Figure 5), we ensure a more comprehensive evaluation of scenarios.

For **UniDS**, we choose one attribute from three options, yielding three settings. In **ConDS**, we address two combinations: (SC, LDD), (SC, UDS), and (LDD, UDS). For each combination, one attribute is selected for the first DS and another for the second, following a $_3C_2$ selection method. For the three-attribute combination in **ConDS**, namely (SC, LDD, UDS), one attribute is chosen for the first DS, another for the second, and the remaining for the last DS, in line with a $_3C_3$ approach. With five datasets in a controlled setup, this generates a total of 165 (source, target) pairs.

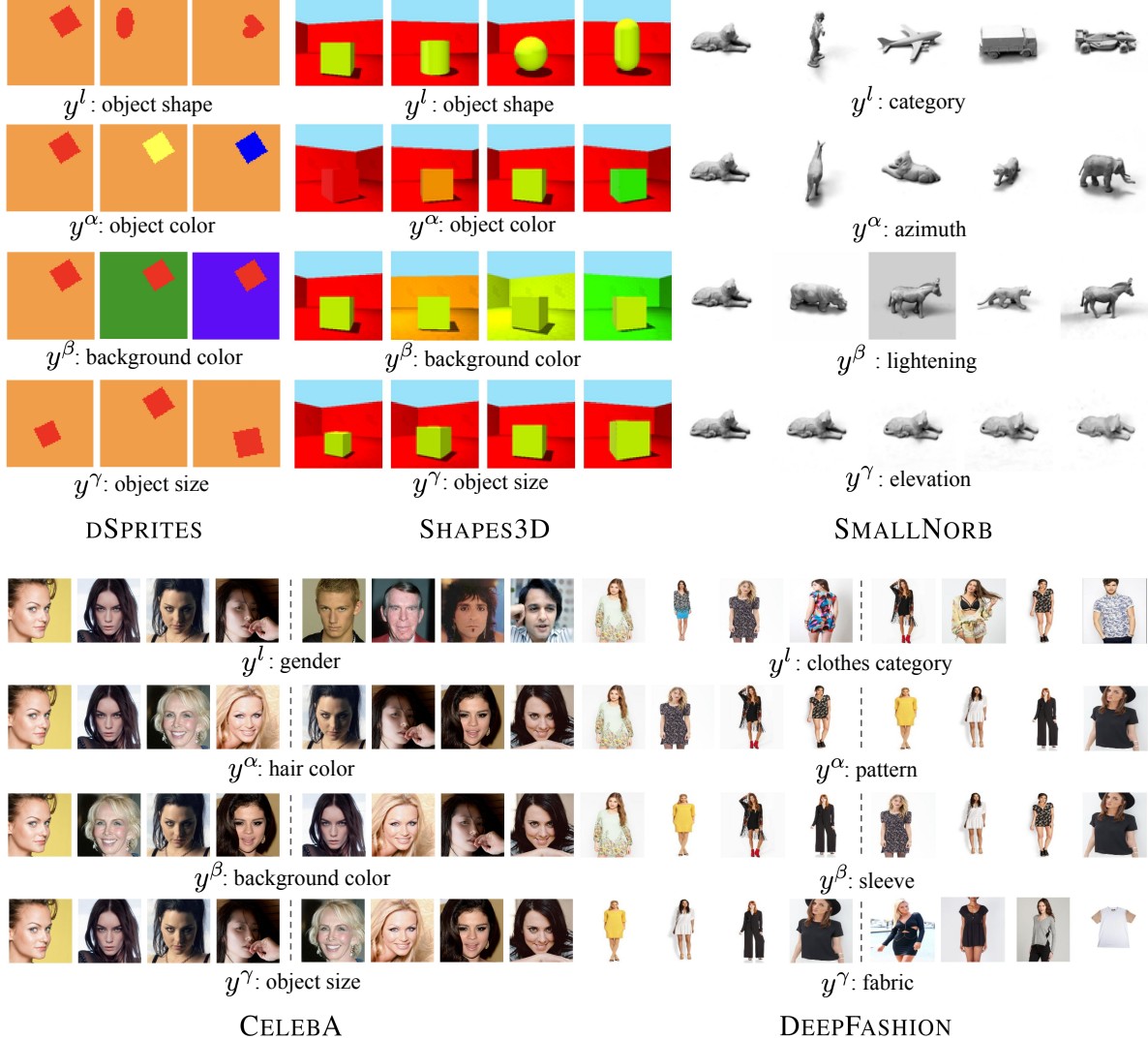

Figure 5: **Samples of controlled datasets.** We provide visualizations of some samples along with their attributes.

## A.2 Dataset Configuration

We provide detailed counts for each split within the datasets. To create the DSs as outlined in Section 3, variations in dataset sizes across the DSs result from the limited availability of certain attribute combinations.

Table 6: **Dataset size.** Please note that we have included 1% counterexamples for SC in our input.

|  | dSprites | Shapes3D | SmallNorb | CelebA | DeepFashion |
|---|---|---|---|---|---|
| *UNIFORM* | 1,091 | 65 | 127 | 227 | 97 |
| *SC* | 1,091 | 65 | 127 | 227 | 97 |
| *LDD* | 1,080 | 624 | 1,600 | 224 | 112 |
| *UDS* | 1,080 | 192 | 500 | 224 | 96 |
| *SC + LDD* | 1,091 | 158 | 324 | 227 | 99 |
| *SC + UDS* | 1,091 | 49 | 101 | 227 | 101 |
| *LDD + UDS* | 1,080 | 468 | 1,280 | 224 | 98 |
| *SC+ LDD + UDS* | 1,091 | 119 | 259 | 227 | 92 |
| *TEST* | 810 | 1,280 | 3,125 | 800 | 80 |

Additionally, we detail the method used to introduce distribution shifts in the original datasets as follows:

- **dSprites:** There are no predefined splits, such as train, validation, or test, in the dSprites dataset; it contains only images and their attribute information. We first created train and test pools by randomly splitting the images, then sampled from each pool to introduce distribution shifts. For the object size attribute, we used the attribute values $[0.8, 0.9, 1]$, labeling them as small, medium, and large, respectively. Since the original dataset is grayscale, we applied our own colorization: Object colors include red $(255, 0, 0)$, yellow $(255, 255, 0)$, and blue $(0, 0, 255)$, while background colors are set as orange $(255, 153, 51)$, green $(0, 153, 0)$, and purple $(102, 0, 255)$. We allocated 20% of the training set as the validation set for parameter tuning, ensuring that both share the same distribution.

- **Shapes3D:** There are no predefined splits, such as train, validation, or test, in the Shapes3D dataset; it contains only images and their attribute information. For the object size attribute, we used the attribute values $[0.75, 0.964, 1.036, 1.179]$, labeling them as tiny, small, medium, and large, respectively. We used $[0, 0.1, 0.2, 0.3]$ for object color and $[0, 0.1, 0.2, 0.3]$ for background color. For each unique combination of labels and attributes, we divided the data instances into training and testing sets. We allocated 20% of the training set as the validation set for parameter tuning, ensuring that both share the same distribution.

- **SmallNorb:** We used the original train-test split provided by SmallNorb, selecting elevations $[0, 2, 4, 6, 8]$, azimuths $[0, 8, 16, 24, 32]$, and lighting conditions $[0, 1, 2, 3, 4]$. To ensure consistency across categories, we manually adjusted the azimuth values for all animal categories, as their initial starting points (0) differed. We allocated 20% of the training set as the validation set for parameter tuning, ensuring that both share the same distribution.

- **CelebA:** Using all samples in the dataset, we first selected the attributes *Black_Hair*, *Smiling*, and *Straight_Hair*. We then split each combination of these attributes into training and testing sets. The training set was used to create distribution shifts, while all test splits were combined to form a uniform distribution. These three attributes were chosen because they provide sufficient samples to accommodate distribution shifts. We allocated 15% of the training set as the validation set for parameter tuning, ensuring that both share the same distribution.

- **DeepFashion:** Using all samples in the dataset, we first selected the attributes *texture* {floral, solid}, *shape* {mini_length, no_dress}, and *style* {chiffon, cotton}. We then split each combination of these attributes into training and testing sets. The training set was used to create distribution shifts, while all test splits were combined to form a uniform distribution. These three attributes were chosen because they provide sufficient samples to accommodate distribution shifts. We allocated 20% of the training set as the validation set for parameter tuning, ensuring that both share the same distribution.

### A.3 Synthetic and Real-world Datasets

We can precisely manage distribution shifts with synthetic datasets because they allow for complete control. Real-world datasets, on the other hand, offer the advantage of realism but may experience uncontrolled distribution shifts. For instance, suppose we create a gender classifier with a dataset that exhibits a spurious correlation by sampling images of older females and younger males. Despite our efforts, we cannot ensure uniformity across other attributes such as skin tone and gender. Given the advantages and disadvantages of each, we conducted experiments with a variety of synthetic datasets (**Dsprites**, **Shapes3D**, **Smallnorb**) and realistic datasets (**CelebA**, **DeepFashion**, **iWildCam**, **fMoW**, **Camelyon17**).

### A.4 Benchmarking Methodology

We outline the benchmarking standards in Table 7.

Table 7: **Benchmarking Methodology**

| Aspect | Configuration | Justification |
|---|---|---|
| **Evaluation Metric** | Accuracy: For the controlled dataset, accuracy is used as the evaluation metric since the test set is uniformly distributed, providing a reliable measure of generalization. For iWildS datasets, accuracy follows the setup of the original paper. | Ensures consistent generalization measurement and alignment with benchmark standards. |
| **Image Standardization** | Pixel values normalized to a range of (-1, 1). | Stabilizes learning and prevents gradient vanishing. |
| **Learning Stop Criterion** | Early stopping applied with a patience of 20 epochs or a maximum of 10,000 iterations. Validation accuracy is measured every 100 iterations; if the highest validation accuracy is not reached, patience decreases by one. Upon achieving the highest validation accuracy, patience resets to 20. | Mitigates overfitting while optimizing model performance. |
| **Image Size** | Configured per dataset: 64x64 (dsprites, shapes3d), 96x96 (smallnorb, camelyon17), 128x128 (deepfashion), 224x224 (fmow, iwildcam), and 256x256 (celeba). | We maintain the original image sizes of the datasets. |

### A.5 Computing Resource

All our experiments were performed using 8 NVIDIA H100 80GB HBM3 GPUs and Intel(R) Xeon(R) Gold 6448Y.

## A.6 Implementation Details

To generate all the results reported in this script, we fine-tuned the hyperparameters. For the controlled datasets, we adopt early stopping where training stops early once the patience limit is reached. Validation accuracy is measured every 100 iterations and one patience is consumed if the best validation accuracy does not improve. The specific values are detailed in Table 8 and Table 9. We conducted a grid search with these parameters to optimize results for each algorithm across all DSs and datasets.

Table 8: **General Hyperparameters.**

| Dataset | Learning Rate | Batch Size | Duration | Patience | Optimizer |
|---|---|---|---|---|---|
| **Controlled** | [1e-2, 1e-3, 1e-4] | 128 | 10000 iterations | 20 | Adam |
| **Real-world** | [1e-3, 5e-4, 1e-4, 5e-5, 1e-5] | 128 | 10000 iterations | 20 | Adam |

Table 9: **Hyperparameters for specific methods.**

| Method | Dataset | Hyperparameters |
|---|---|---|
| ViT | All | Model: ViT_B_16 |
| MLP | All | #layers: 4, hidden dim: 256 |
| ImageNet | **Controlled** | scale lower bound: 0.08 |
|  | **Real-world** | scale lower bound: 0.3 |
| AugMix | **Controlled** | severity: 3, mixture width: 3 |
|  | **Real-world** | severity: 7, mixture width: 5 |
| RandAug | **Controlled** | num_ops: 3, magnitude: 5 |
|  | **Real-world** | num_ops: 2, magnitude: 9 |
| AutoAug | All | CIFAR10 policy for **dSprites**, **Shapes3D**, and **SmallNorb**. |
|  |  | Otherwise, ImageNet policy |
| UBNet | All | base model patience: 10, base model training epochs: 50 |
| PnD | All | $\alpha_1$: 0.2, $\alpha_2$: 2, $\beta$: 4, $q$: 0.7, $\text{EMA}_\alpha$: 0.7, |
|  |  | base model patience: 10, base model training epochs: 50 |
| OccamNets | All | $\lambda_{CS}$: 0.1, $\lambda_G$: 1, $\gamma_0$: 3, $\gamma$: 1, $\tau_{acc,0}$: 0.5 |
|  | **fMoW**: | $\tau_{acc,0}$: 0.2 |
| GroupDRO | All | $\lambda$: 1e-2 |
| BPA | All | $k$: 8, $m$: 0.3, base model patience: 10, base model training epochs: 50 |
|  | **iWildcam** | $k$: 12 |
| ADA | All | $T_{max}$: 15, $k$: 2, $\gamma$: 1.0 |
| ME-ADA | All | $T_{max}$: 15, $k$: 2, $\gamma$: 1, $\eta$: 1.0 |
| SagNet | All | $\lambda_{adv}$: 0.1 |
| L2D | All | $\alpha_1$: 1, $\alpha_2$: 1, $\beta$: 0.1 |
| IRM | All | annealing iterations: 500, $\lambda$: 1 (during annealing), 100 (after annealing) |
| CausIRL | **Controlled** | $\gamma$: 0.5 (**SmallNorb**), 0.3 (**dSprites**), 0.1 (**DeepFashion**, **CelebA**), 1 (**Shapes3D**) |
|  | **Real-world** | $\gamma$: 1 |
| CLIP-base | All | model: "openai/clip-vit-base-patch32" |
| CLIP-Large | All | model: "openai/clip-vit-large-patch14" |
| InstructBLIP | Inference | model: "Salesforce/instructblip-vicuna-7b",num_beam: 5, load_in_4bit: True |
|  |  | top_p: 0.9, repetition_penalty: 1.5, length_penalty: 1.0, temperature: 1, max_new_tokens: 20 |
|  | Fine-tuning | lora-r: 8 |
| LLaVA-1.5 | Inference | model: "llava-hf/llava-1.5-7b-hf", max_new_tokens: 200 |
|  | Fine-tuning | lora-r: 128 |
| Phi-3.5-vision | Inference | model: "microsoft/Phi-3.5-vision-instruct", max_new_tokens: 200, temperature: 0.2 |
|  | Fine-tuning | lora-r: 64 |
| GPT-4o mini | All | temperature:1, $top_p$:1 |
| GPT-4o | All | temperature:1, $top_p$:1 |

# B Experimental Results

## B.1 Comprehensive Results

Figure 6 displays the results for all algorithms trained from scratch across all DSs, while Figure 7 presents the outcomes for all algorithms trained using ImageNet pre-trained weights.

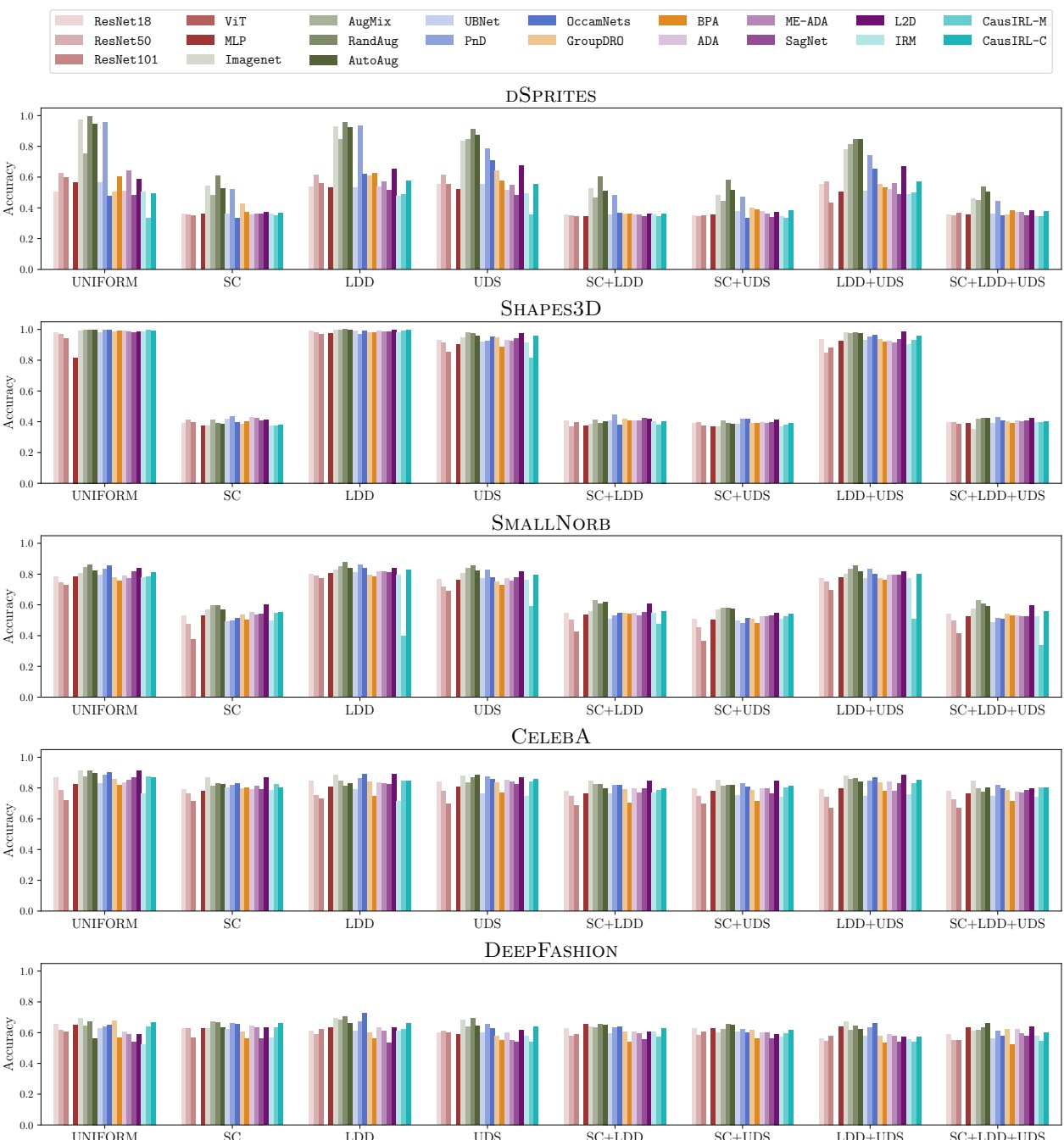

Figure 6: **Results for all algorithms from scratch.** We plot the top-1 accuracy.

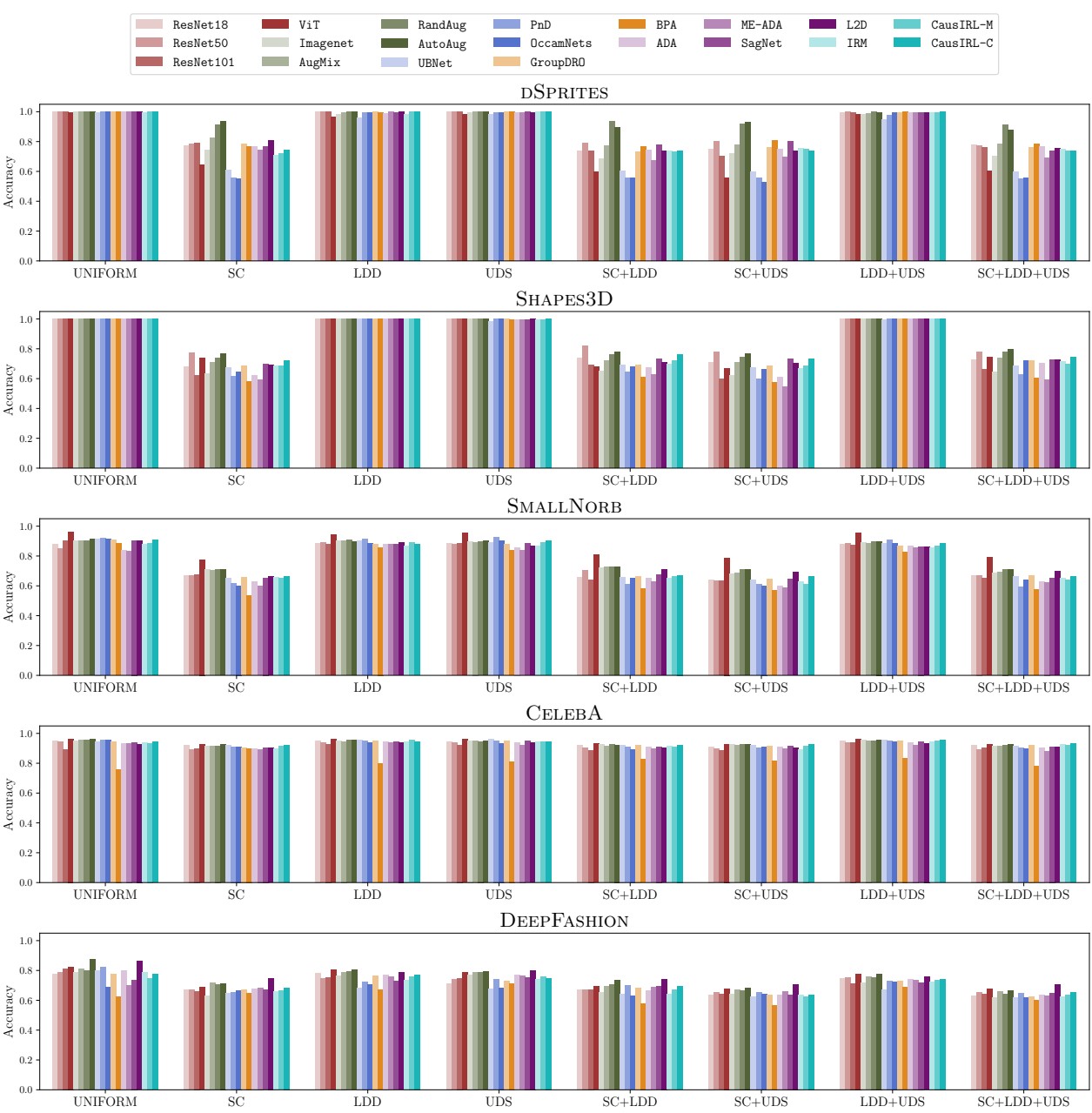

Figure 7: **Results for all algorithms with pre-trained weight.** We plot the top-1 accuracy.

## B.2 Experimental Study on Pre-training

Figure 8 presents the aggregate results of pre-training across all algorithms and DSs.

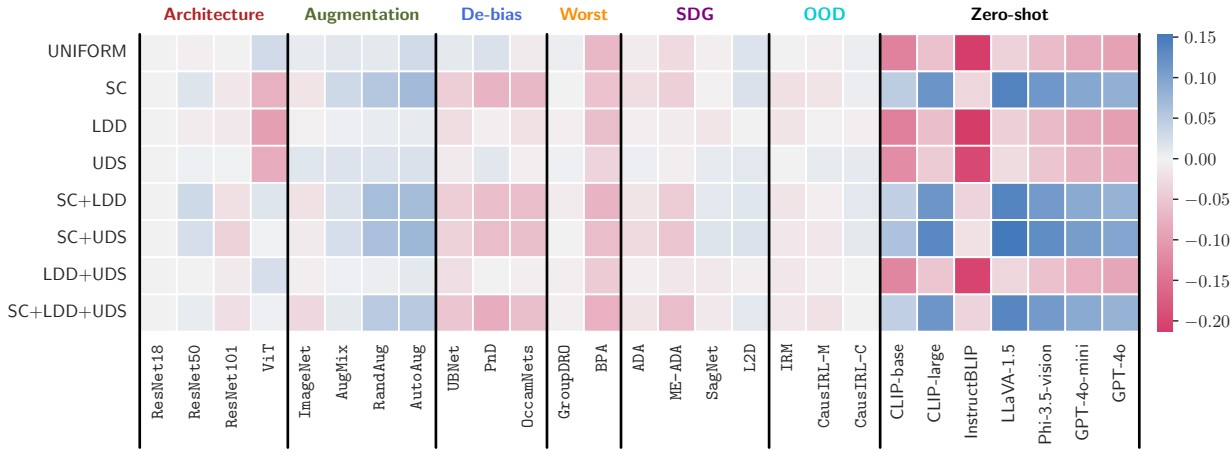

Figure 8: **Result with ImageNet pre-trained weight**. The setup is the same as Figure 3.

## B.3 Analysis across Different Data Sizes

We further analyze the aggregate results across different dataset sizes. As shown in Table 10, we evaluate the algorithms in three variations: small, medium, and large (see from Figure 9 to Figure 20). Although the overall trend remains similar, some differences are observed.

Table 10: **Various Dataset size.** Please note that we have included 1% counterexamples for SC in our input.

|  | Shapes3D | | | CelebA | | |
|---|---|---|---|---|---|---|
|  | small | middle | big | small | middle | big |
| SC | 65 | 130 | 194 | 114 | 227 | 340 |
| LDD | 624 | 1,248 | 1,872 | 112 | 224 | 336 |
| UDS | 192 | 384 | 576 | 112 | 224 | 336 |
| SC + LDD | 158 | 316 | 473 | 114 | 227 | 340 |
| SC + UDS | 49 | 97 | 146 | 114 | 227 | 340 |
| LDD + UDS | 468 | 936 | 1,404 | 112 | 224 | 336 |
| SC+ LDD + UDS | 119 | 237 | 355 | 114 | 227 | 340 |

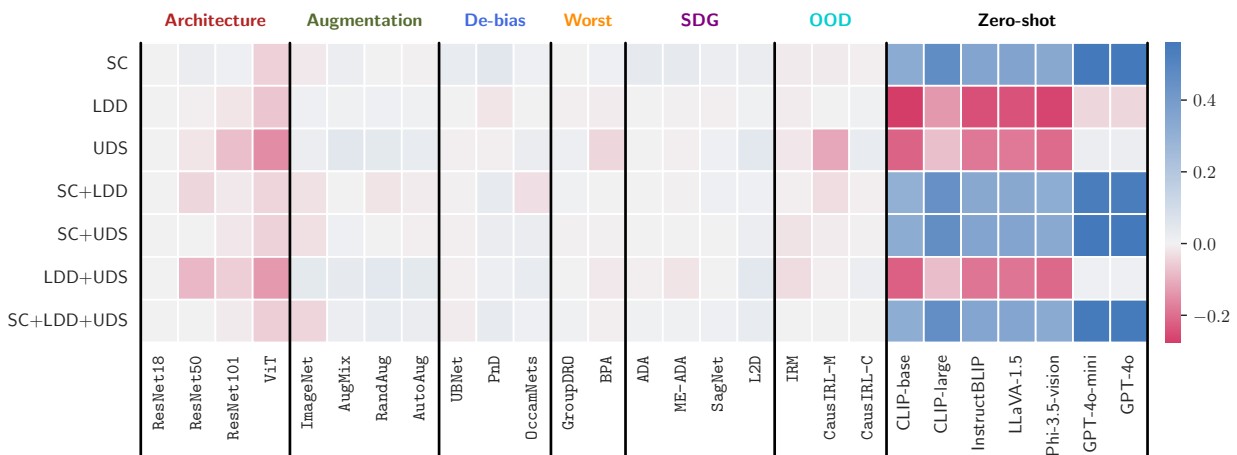

Figure 9: **Scratch Shapes3D result with small dataset size**. The setup is the same as Figure 3.

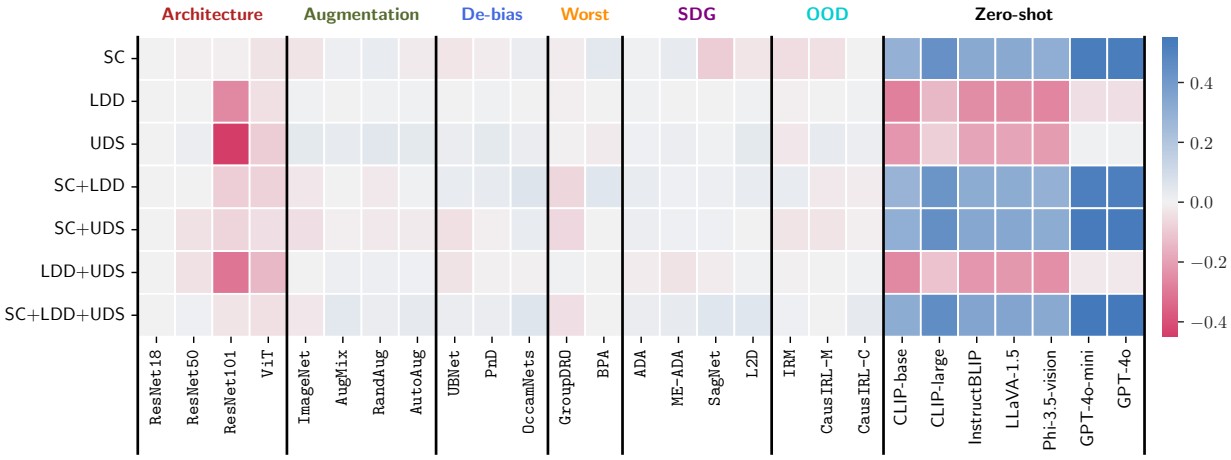

Figure 10: **Scratch Shapes3D result with middle dataset size**. The setup is the same as Figure 3.

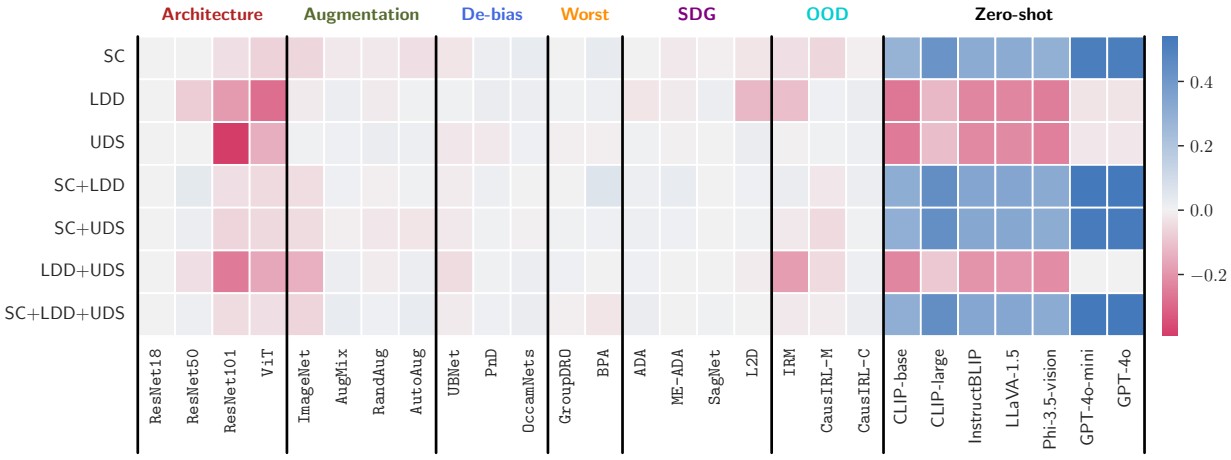

Figure 11: **Scratch Shapes3d result with big dataset size**. The setup is the same as Figure 3.

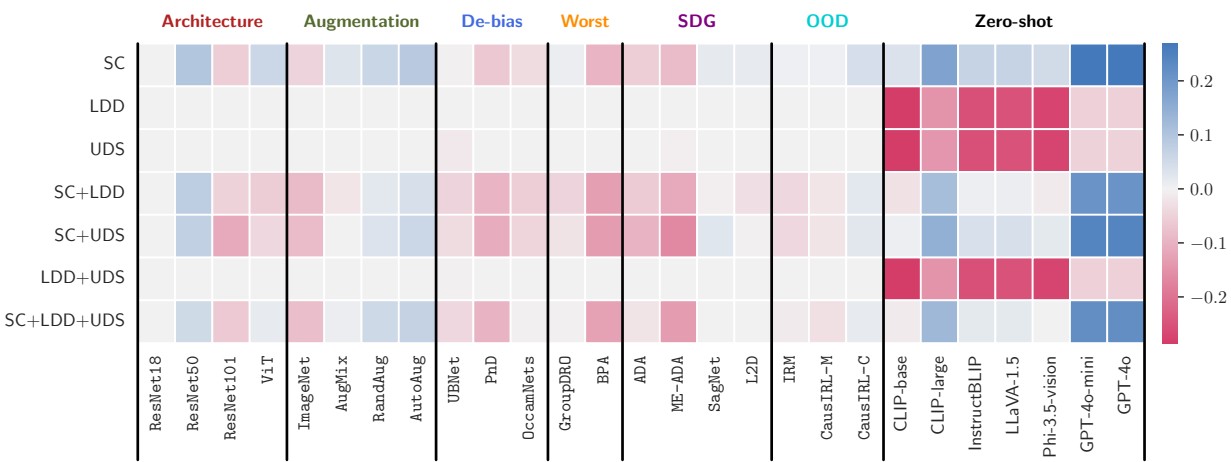

Figure 12: **Pretrain Shapes3d result with small dataset size**. The setup is the same as Figure 3.

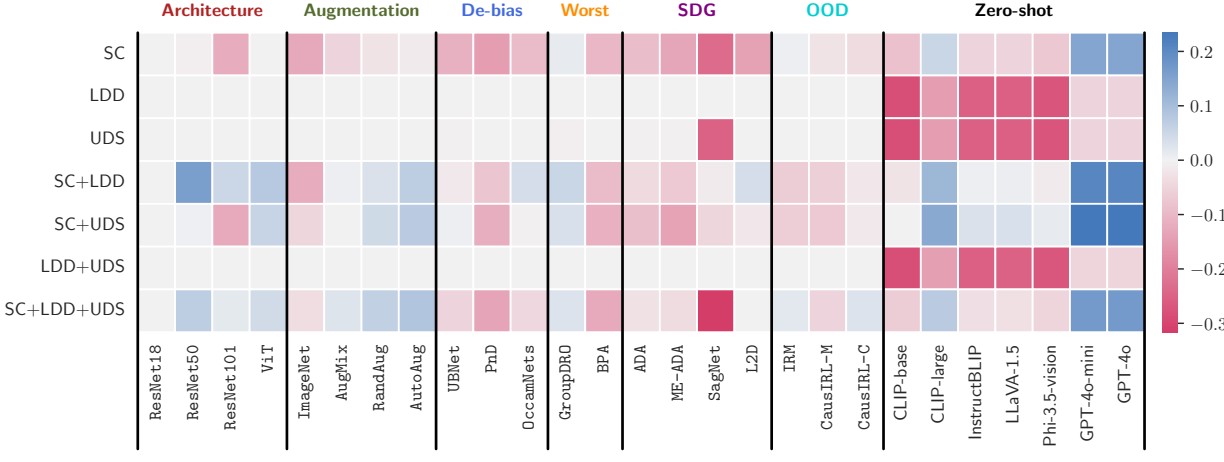

Figure 13: **Pretrain Shapes3d result with middle dataset size**. The setup is the same as Figure 3.

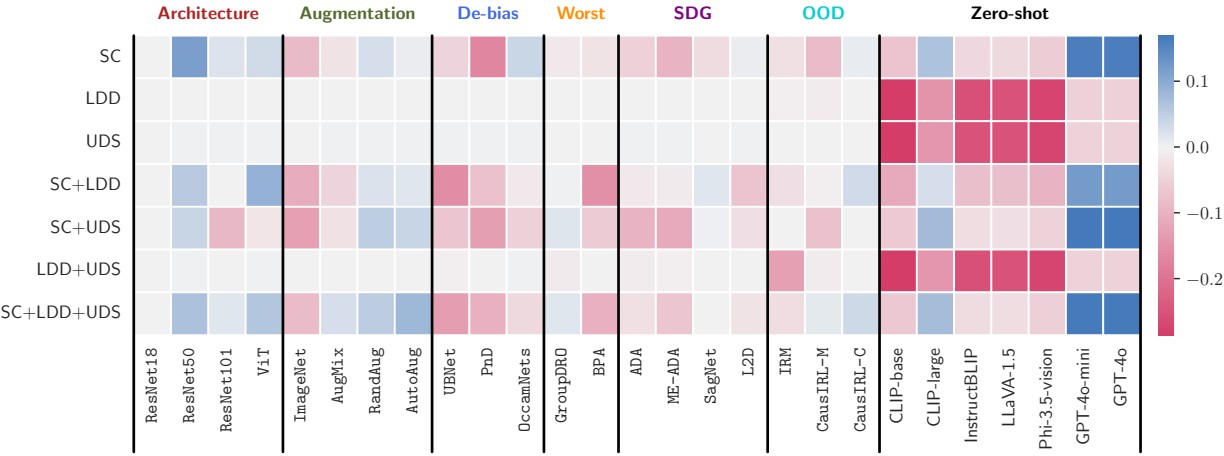

Figure 14: **Pretrain Shapes3d result with big dataset size**. The setup is the same as Figure 3.

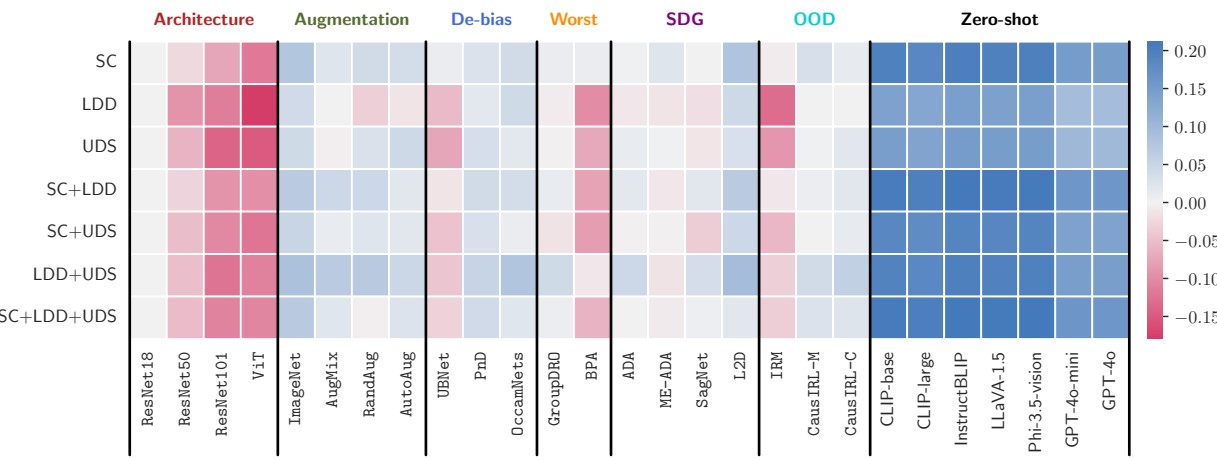

Figure 15: **Scratch CelebA result with small dataset size**. The setup is the same as Figure 3.

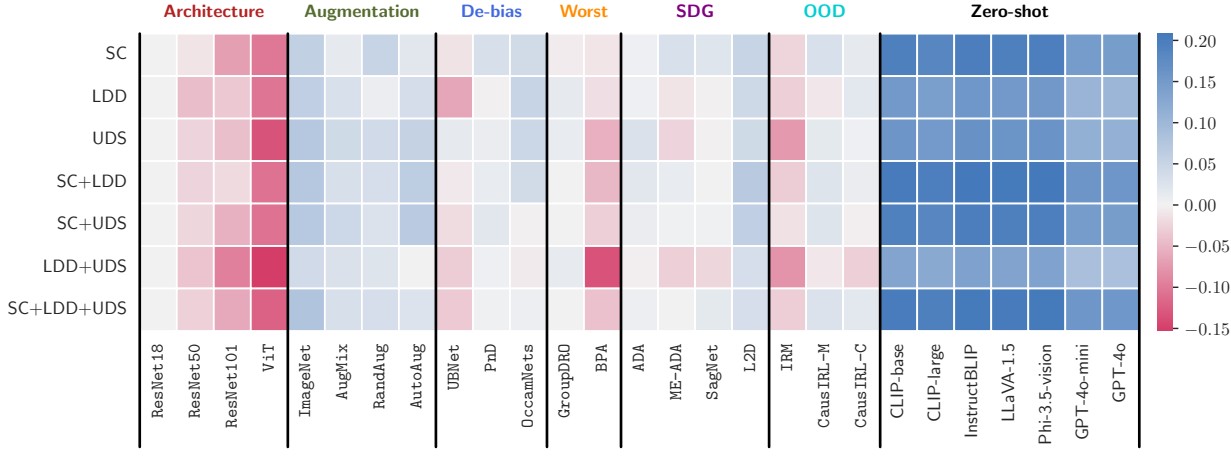

Figure 16: **Scratch CelebA result with middle dataset size**. The setup is the same as Figure 3.

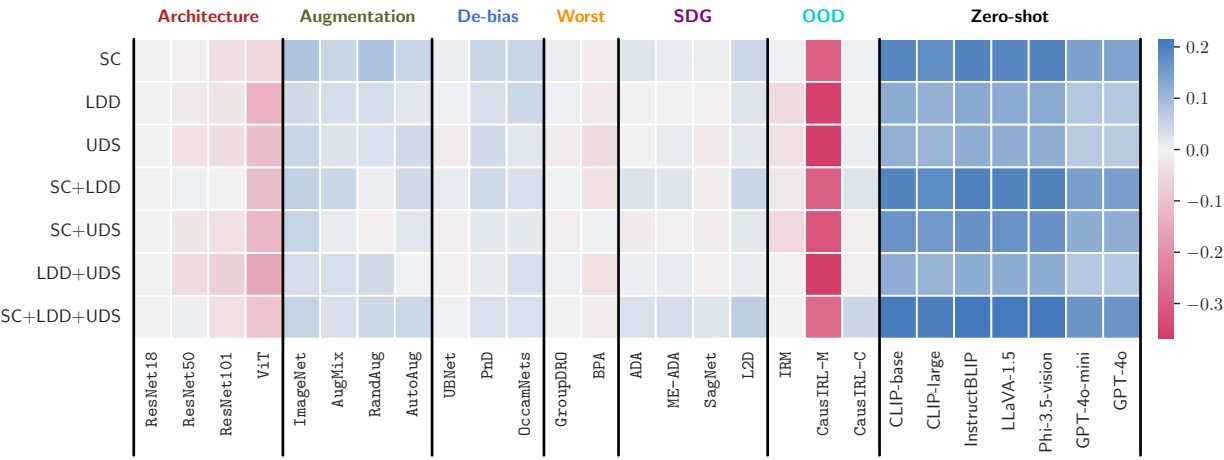

Figure 17: **Scratch CelebA result with big dataset size**. The setup is the same as Figure 3.

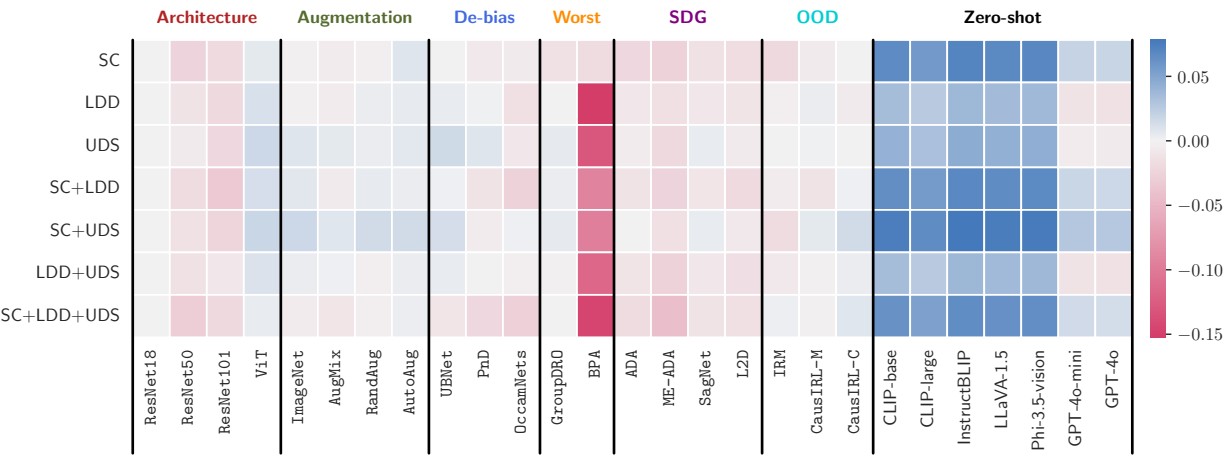

Figure 18: **Pretrain CelebA result with small dataset size**. The setup is the same as Figure 3.

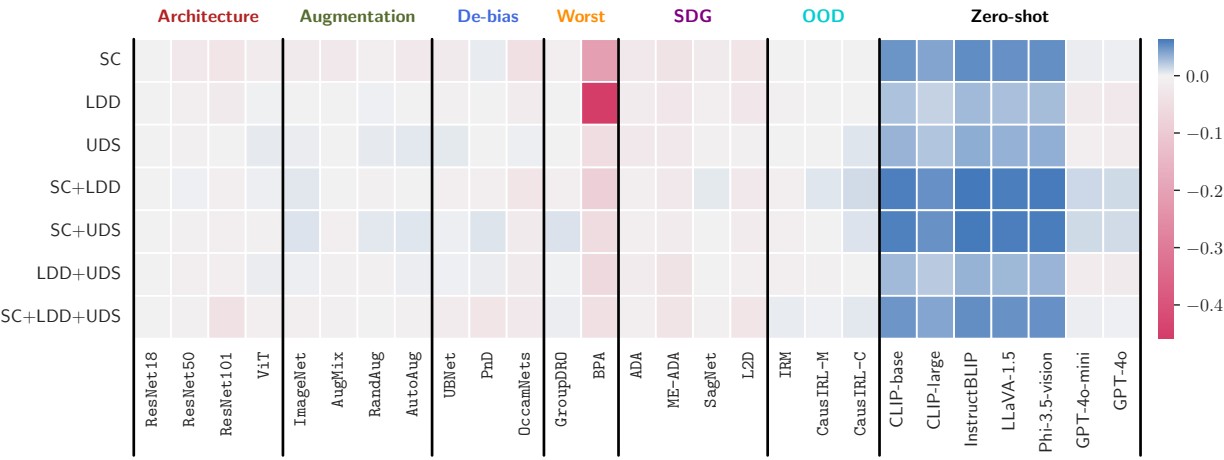

Figure 19: **Pretrain CelebA result with middle dataset size**. The setup is the same as Figure 3.

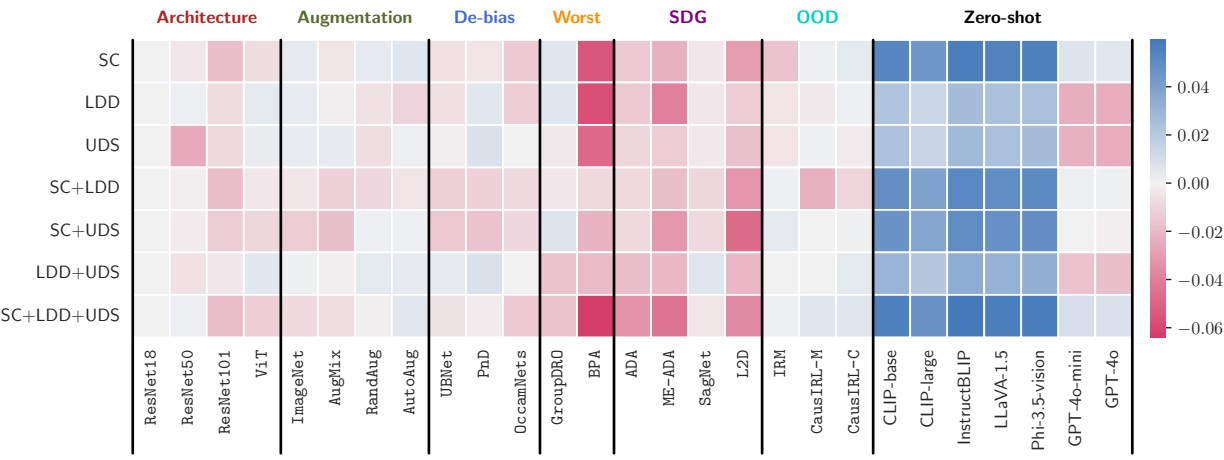

Figure 20: **Pretrain CelebA result with big dataset size**. The setup is the same as Figure 3.

## B.4 Results for Real-world datasets

Table 11 presents the detailed results for each dataset listed in Table 3.

Table 11: **Experimental results on real-world datasets.** The setting is exactly the same as Table 3.

| | | iWildCam | | fMoW | | Camelyon17 | |
|---|---|---|---|---|---|---|---|
| | | *Scratch* | *Pre-training* | *Scratch* | *Pre-training* | *Scratch* | *Pre-training* |
| *Architecture* | ResNet18 | 51.77(±0.47) | 59.50(±0.43) | 27.13(±0.44) | 41.08(±0.44) | 82.83(±0.25) | 84.35(±0.10) |
| | ResNet50 | 53.35(±0.34) | 66.75(±0.39) | 28.55(±0.44) | 49.75(±0.32) | 75.85(±0.36) | 86.78(±0.38) |
| | ResNet101 | 43.41(±0.07) | 69.71(±0.20) | 22.35(±0.30) | 48.63(±0.17) | 75.26(±0.37) | 84.21(±0.49) |
| | ViT | 51.68(±0.33) | 69.82(±0.45) | 23.20(±0.53) | 51.74(±0.46) | 76.78(±0.18) | 91.27(±0.34) |
| | *avg.* | *50.05(±2.25)* | *66.45(±2.42)* | *25.31(±1.50)* | *47.80(±2.33)* | *77.68(±1.74)* | *86.65(±1.65)* |
| *Augmentation* | Imagenet | 48.21(±0.21) | 64.17(±0.39) | 26.16(±0.53) | 47.32(±0.33) | 72.08(±0.34) | 87.95(±0.39) |
| | AugMix | 51.90(±0.29) | 65.61(±0.43) | 29.94(±0.27) | 49.95(±0.21) | 74.13(±0.41) | 88.72(±0.49) |
| | RandAug | 47.49(±0.28) | 65.14(±0.05) | 31.12(±0.24) | 49.48(±0.48) | 77.21(±0.41) | 90.24(±0.40) |
| | AutoAug | 51.57(±0.32) | 70.19(±0.03) | 31.64(±0.44) | 50.13(±0.24) | 73.23(±0.28) | 90.56(±0.47) |
| | *avg.* | *49.79(±1.13)* | *66.28(±1.34)* | *29.71(±1.24)* | *49.22(±0.65)* | *74.16(±1.10)* | *89.37(±0.62)* |
| *De-biasing* | UBNet | 32.96(±0.47) | 57.96(±0.39) | 16.30(±0.48) | 37.06(±0.51) | 73.77(±0.38) | 82.06(±0.12) |
| | PnD | 45.25(±0.42) | 61.03(±0.48) | 21.01(±0.54) | 40.29(±0.40) | 83.81(±0.40) | 79.69(±0.28) |
| | OccamNets | 53.67(±0.49) | 67.30(±0.45) | 29.08(±0.27) | 46.53(±0.39) | 74.97(±0.32) | 81.77(±0.47) |
| | *avg.* | *43.96(±6.01)* | *62.10(±2.75)* | *22.13(±3.73)* | *41.29(±2.78)* | *77.52(±3.17)* | *81.17(±0.75)* |
| *Worst-case* | GroupDRO | 45.15(±0.22) | 63.15(±0.23) | 25.99(±0.46) | 49.07(±0.44) | 73.37(±0.38) | 80.99(±0.35) |
| | BPA | 35.54(±0.23) | 56.06(±0.18) | 8.71(±0.26) | 44.32(±0.17) | 76.99(±0.27) | 86.08(±0.14) |
| | *avg.* | *40.35(±4.85)* | *59.60(±3.54)* | *17.35(±8.64)* | *46.70(±2.38)* | *75.18(±1.81)* | *83.53(±2.55)* |
| *SDG* | ADA | 48.47(±0.41) | 69.99(±0.45) | 29.71(±0.36) | 50.40(±0.41) | 80.45(±0.20) | 88.74(±0.25) |
| | ME-ADA | 52.62(±0.26) | 65.87(±0.40) | 28.46(±0.33) | 51.43(±0.35) | 80.29(±0.19) | 81.93(±0.03) |
| | SagNet | 54.67(±0.50) | 69.83(±0.38) | 29.53(±0.33) | 49.32(±0.32) | 78.62(±0.39) | 81.10(±0.45) |
| | L2D | 47.78(±0.16) | 64.09(±0.49) | 23.90(±0.30) | 45.65(±0.34) | 86.32(±0.23) | 94.85(±0.27) |
| | *avg.* | *50.88(±1.65)* | *67.44(±1.47)* | *27.90(±1.36)* | *49.20(±1.26)* | *81.42(±1.68)* | *86.66(±3.22)* |
| *OOD* | IRM | 34.55(±0.51) | 57.36(±0.31) | 11.40(±0.51) | 30.88(±0.36) | 72.47(±0.35) | 85.45(±0.41) |
| | CausIRL-M | 52.28(±0.50) | 68.34(±0.45) | 22.32(±0.28) | 49.44(±0.11) | 72.10(±0.26) | 79.83(±0.14) |
| | CausIRL-C | 53.36(±0.40) | 65.36(±0.47) | 28.61(±0.18) | 49.83(±0.13) | 81.16(±0.21) | 88.98(±0.37) |
| | *avg.* | *46.73(±6.10)* | *63.69(±3.28)* | *20.78(±5.03)* | *43.38(±6.25)* | *75.24(±2.96)* | *84.75(±2.66)* |
| *Zero-shot* | CLIP-base | 13.97 | | 13.30 | | 50.01 | |
| | CLIP-large | 13.70 | | 23.46 | | 50.01 | |
| | InstructBLIP | 1.86 | | 18.17 | | 68.03 | |
| | LLaVA-1.5 | 4.64 | | 16.32 | | 51.09 | |
| | Phi-3.5-vision | 9.91 | | 10.88 | | 66.71 | |
| | GPT-4o mini | 43.36 | | 19.89 | | 64.98 | |
| | GPT-4o | 51.80 | | 25.58 | | 58.10 | |
| | *avg.* | *19.89* | | *18.23* | | *58.42* | |

## B.5 Further Analysis on Zero-shot Inference

To utilize foundation models for image classification, we employed various prompts to extract labels from the input images. Table 12 and Table 13 detail the prompts used in our prompt engineering approach.

Table 12: **Prompts used in Zero-shot inference.**

| Foundation Model | Prompt Type | Prompt |
|---|---|---|
| *CLIP* | General | "a photo of a *label*" |
| LLaVA-1.5 InstructBLIP Phi-3.5-vision GPT-4o mini GPT-4o | General$_1$ | Classify the image into $label_1, ...,$ or $label_C$. Please provide only the name of the label. |
| | General$_2$ | Choose a label that best describes the image. Here is the list of labels to choose from: $label_1, ..., label_C$. Please provide only the name of the label. |
| | **dSprites Shapes3D SmallNorb** | Classify the object in the image into $label_1, ...,$ or $label_C$. Please provide only the name of the label. |
| | **CelebA** | Classify the person in the image into $male$ or $female$. Please provide only the name of the label. |
| | **DeepFashion** | Is a person wearing a dress or not? Please answer in yes or no. |
| | **iWildcam** | Classify the object or animal in the image. Here is the list of labels to choose from: $label_1, ..., label_C$. Please provide only the name of the label. |
| | **fMoW** | Classify the building or land-use in the image into $label_1, ..., label_C$. Please provide only the name of the label. |
| | **Camelyon17** General$_1$ | Please answer yes if the image contains any tumor tissue, and no otherwise. Please provide only the name of the label. |
| | **Camelyon17** General$_2$ | Please answer yes if the image contains any tumor tissue, and no otherwise. Please respond with a single word. |
| | **Camelyon17** Tailored | Please analyze the image and determine if it contains any tumor tissue. Respond with 'Yes' if tumor tissue is present, or 'No' if it is not. |

Table 13: **Prompt formats for vision-language models.**

| | *LLaVA-1.5, InstructBLIP* | *Phi-3.5-vision* | *GPT-4o, GPT-4o-mini* |
|---|---|---|---|
| Prompt Format | USER: < image >\n prompt \n ASSISTANT: | USER: < \|image_1\| >\n prompt | prompt |

## B.6 Fine-tuned Open Source Foundation Model

We observe in Table 3 that the zero-shot performance of foundation models is constrained when applied to real-world datasets. Our evaluation on real-world datasets like Camelyon17, which contains complex cell images, iWildCam with its camera trap images of diverse animal species, and FMoW's satellite images present unique challenges due to their niche content and visual complexity. This high level of domain specificity, with features likely outside the foundation models' general scope, limits their capacity to generalize effectively, particularly in zero-shot settings.

An intuitive approach to evaluate this is by fine-tuning the vision encoder on these specialized datasets. Through fine-tuning with LoRA Hu et al. (2022), we found that the foundation models performed as expected, showing high effectiveness for these datasets. The results are presented in Table 14.

Table 14: **Fine-tuned open source foundation model.** w and w/o denote with and without fine-tuning, respectively.

| | iWildCam | | Camelyon17 | | FMoW | |
|---|---|---|---|---|---|---|
| | w/o | w | w/o | w | w/o | w |
| LLaVA-1.5 | 4.64 | **91.12** | 51.09 | **95.32** | 16.32 | **72.67** |
| Phi-3.5-Vision | 9.91 | **91.19** | 66.71 | **93.35** | 10.88 | **77.02** |
| InstructBLIP | 1.86 | **12.13** | 68.03 | **99.87** | 18.17 | **41.18** |

## B.7 Visualization on Invariant Feature Learning.

We investigate feature invariance with respect to labels and attributes using the CelebA dataset. UMAP (McInnes et al., 2018) is utilized for visualization. Figure 21 illustrates the feature space of the best and worst-performing algorithms, while Figure 22 compares learning from scratch with pre-training. While all the algorithms demonstrate invariance to LDD and UDS, ViT exhibits sensitivity to SC. In contrast, both CLIP-Large and ImageNet remain invariant to all DSs. In Figure 22, the ViT model with pre-training exhibits better invariance to attributes compared to the ViT model trained from scratch.

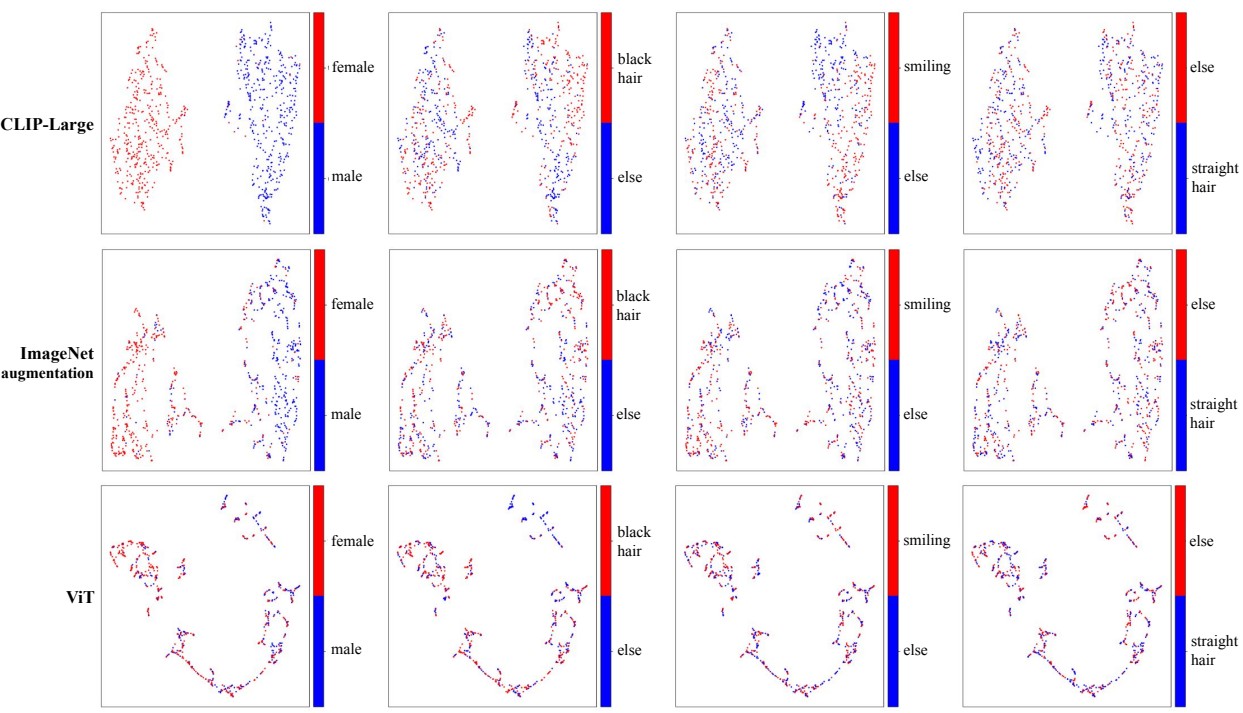

Figure 21: **Visualization on invariant feature learning.** We visualize the top two algorithms, CLIP-Large and ImageNet, along with the worst-performing algorithm. In this setup, 'black hair' is used to create SC, 'smiling' is used to generate LDD, and 'straight hair' is used to produce UDS.

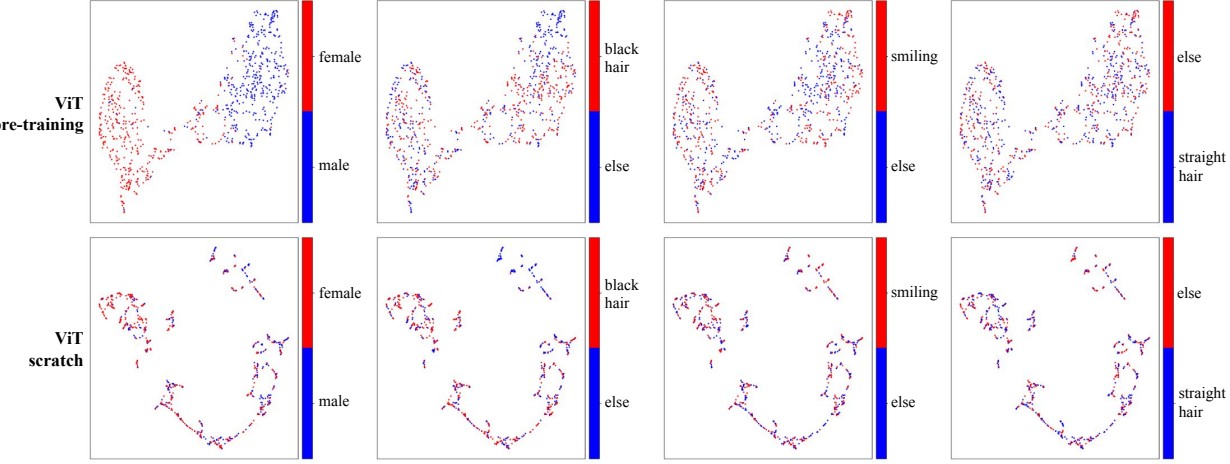

Figure 22: **Scratch vs. Pre-training.** The setup is exactly the same as Figure 21.

## B.8 Generation of Attributes with Augmentations

Our framework requires datasets with rich attribute annotations to create **ConDS**. However, such datasets are limited as annotations are expensive. We did consider using augmentations to create additional attributions, but augmentation techniques in algorithm baselines might directly address these shifts in this setting. However, for the rebuttal, we applied three types of corruptions from ImageNet-C (Hendrycks & Dietterich, 2018)—impulse noise, snow, and elastic transform—on CelebA. Each attribute was evaluated under two conditions (corrupted and uncorrupted). Figures 23 and 24 exhibit the evaluation results with this setup.

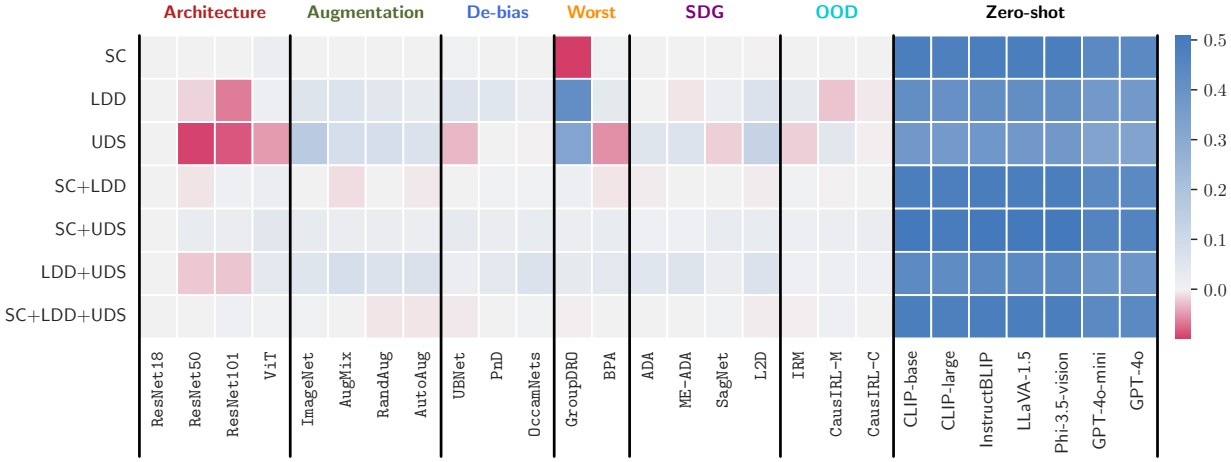

Figure 23: **Scratch CelebA result with attributes generated from augmentation**. The setup is the same as Figure 3.

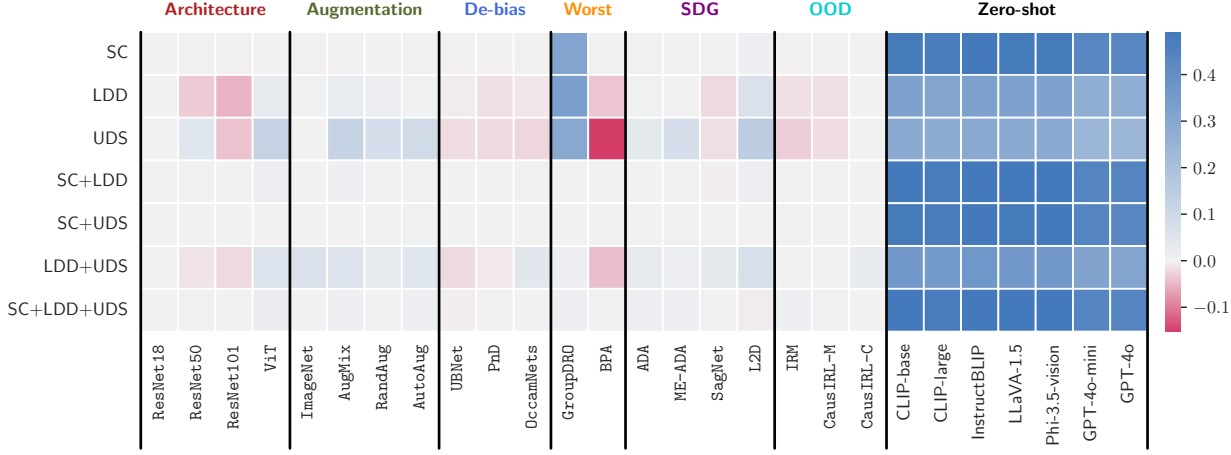

Figure 24: **Pretrain CelebA result with attributes generated from augmentation**. The setup is the same as Figure 3.

### B.9 Results for Distribution Shift Generated by Clustering

We leverage DINOv2 (Oquab et al., 2024) to extract rich image features, using the resulting feature clusters as proxies for DSs. The motivation behind this approach aligns with that of Section B.8, namely, the necessity for a scalable method to simulate DSs without depending on costly and labor-intensive attribute annotations. As displayed in Table 15, we group image features into six clusters based on their embeddings from DINOv2 and treat these clusters as distinct distributions. Figures 25 and 26 exhibit the evaluation results with this setup.

Table 15: **Cluster groups.** From six clusters, we create four distribution shift groups and one test group, simulating different environmental or contextual variations.

|         | Group 1 | Group 2 | Group 3 | Group 4 | Test |
|---------|---------|---------|---------|---------|------|
| *Cluster* | 1, 4  | 2, 4    | 1, 5    | 2, 5    | 3, 6 |

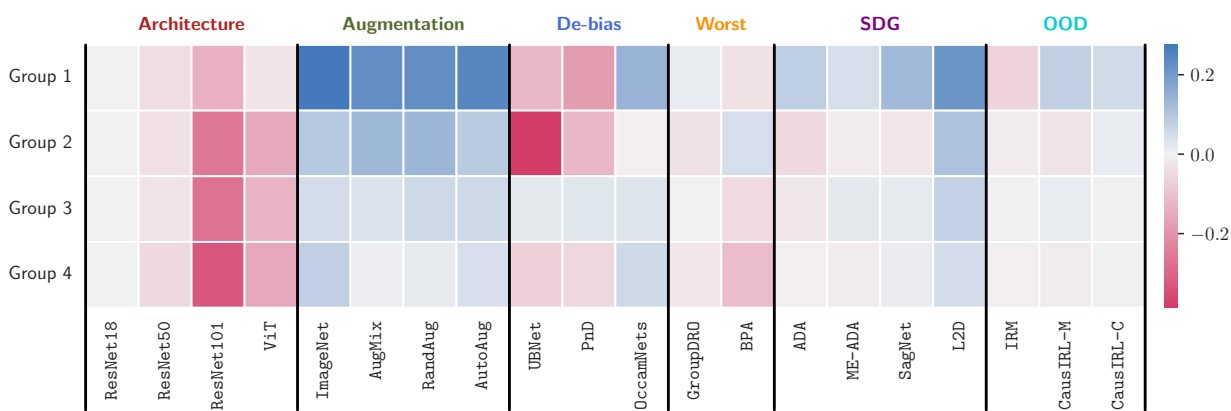

Figure 25: **Scratch CelebA results across cluster groups**. The setup is the same as Figure 3.

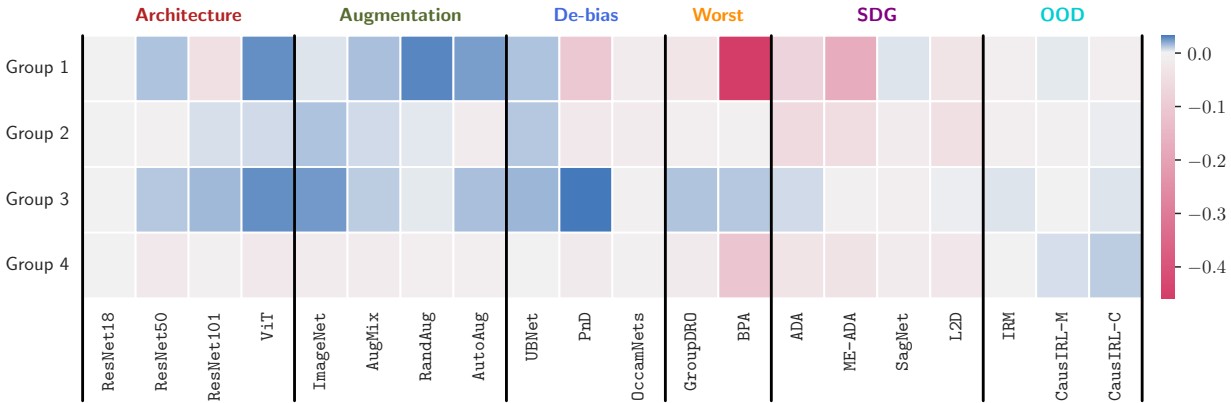

Figure 26: **Pretrain CelebA result across cluster groups**. The setup is the same as Figure 3.

## B.10   Performance Sensitivity for Hyperparameters

We evaluate the performance sensitivity of the algorithms across all datasets and compute the average results. For detailed hyperparameter configurations, please refer to Table 8. Figures 27 - 40 illustrate the results.

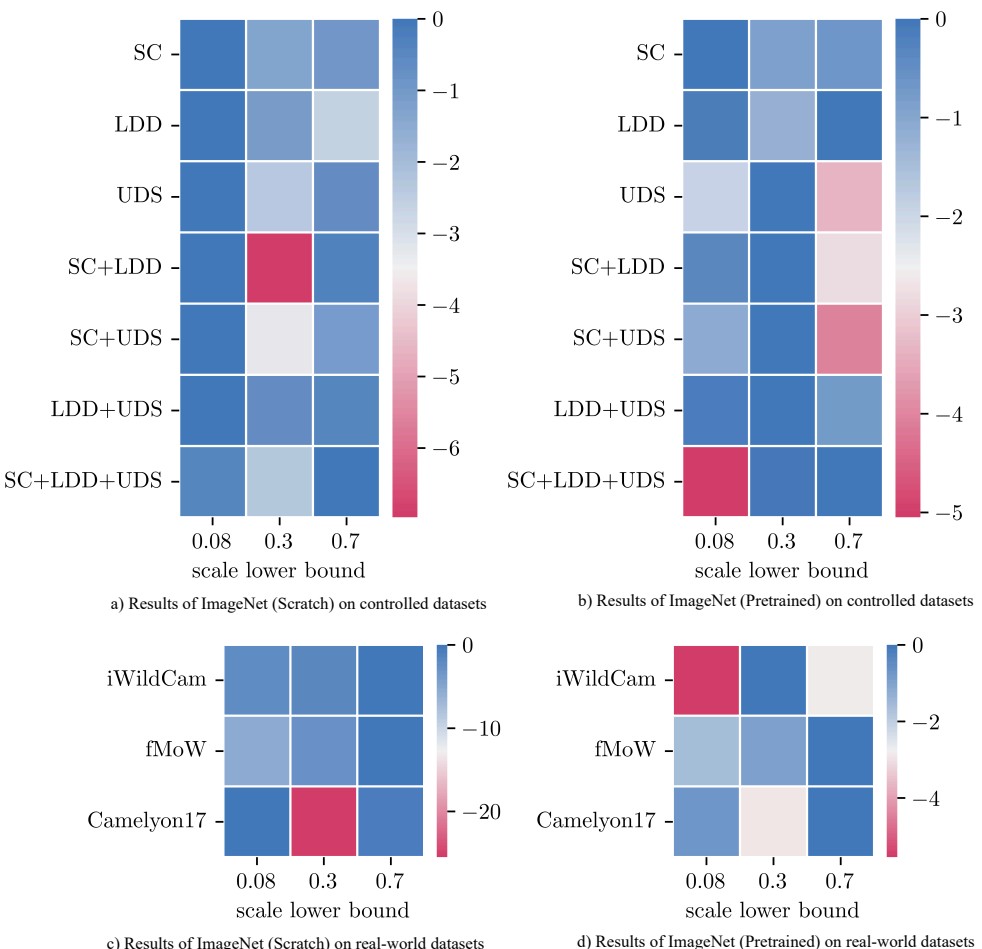

a) Results of ImageNet (Scratch) on controlled datasets

b) Results of ImageNet (Pretrained) on controlled datasets

c) Results of ImageNet (Scratch) on real-world datasets

d) Results of ImageNet (Pretrained) on real-world datasets

Figure 27: **Hyperparameter Exploration for ImageNet**.

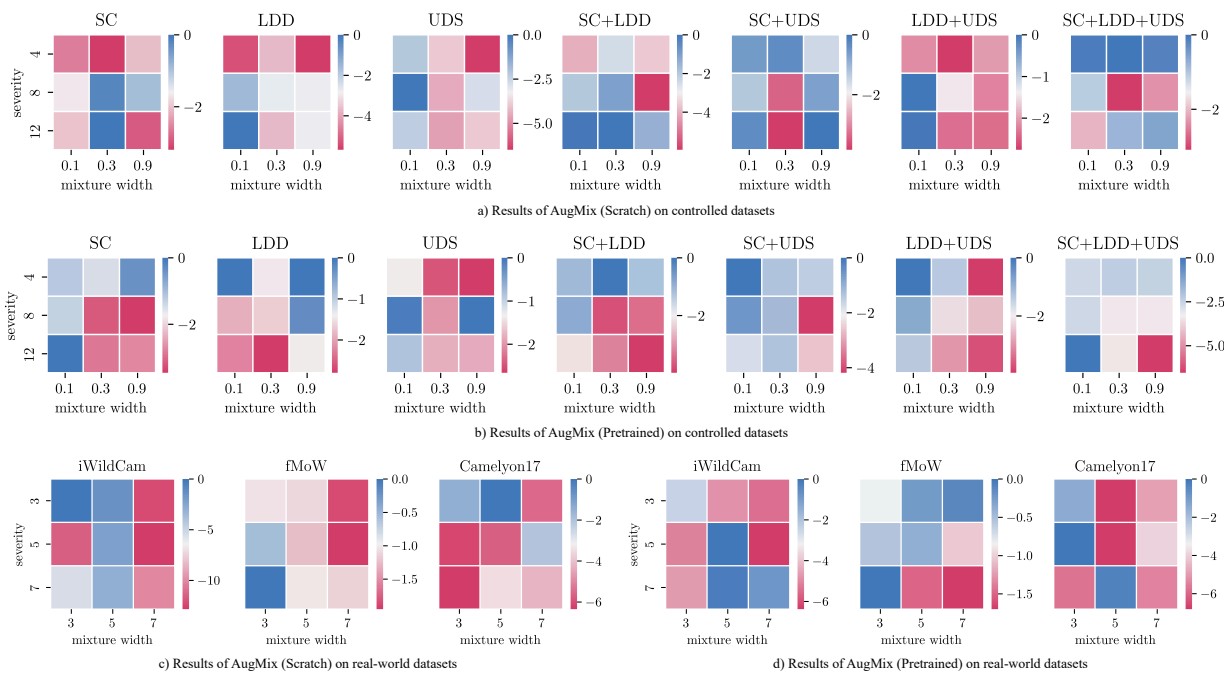

Figure 28: **Hyperparameter Exploration for AugMix**.

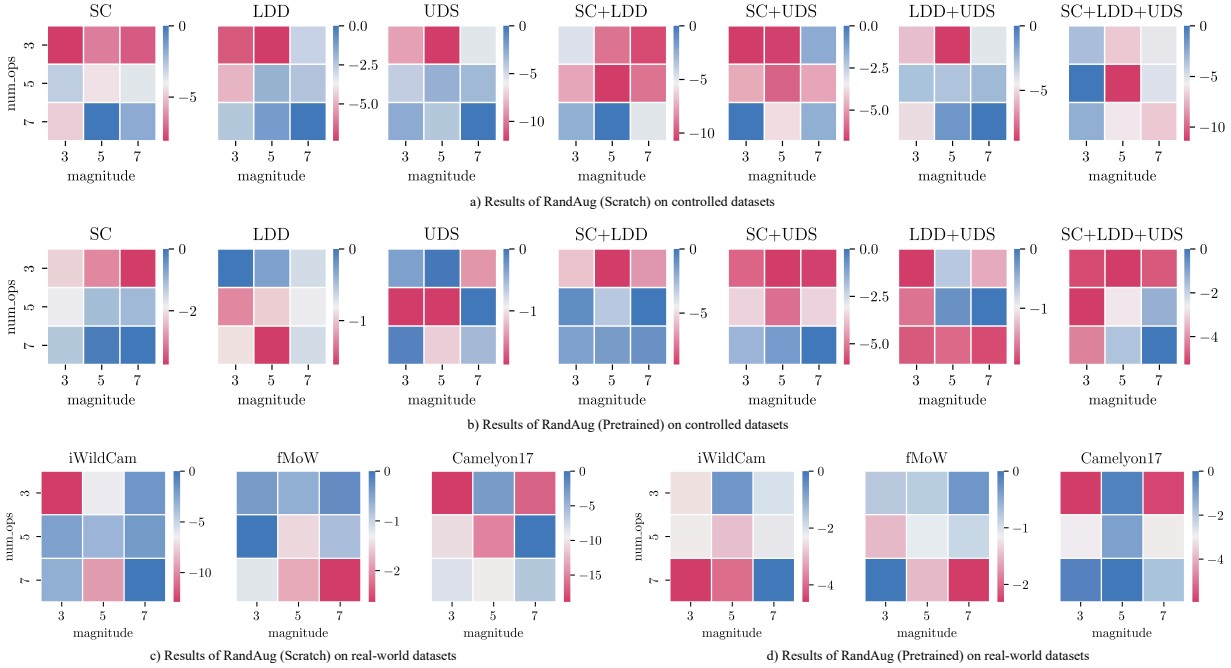

Figure 29: **Hyperparameter Exploration for RandAug**.

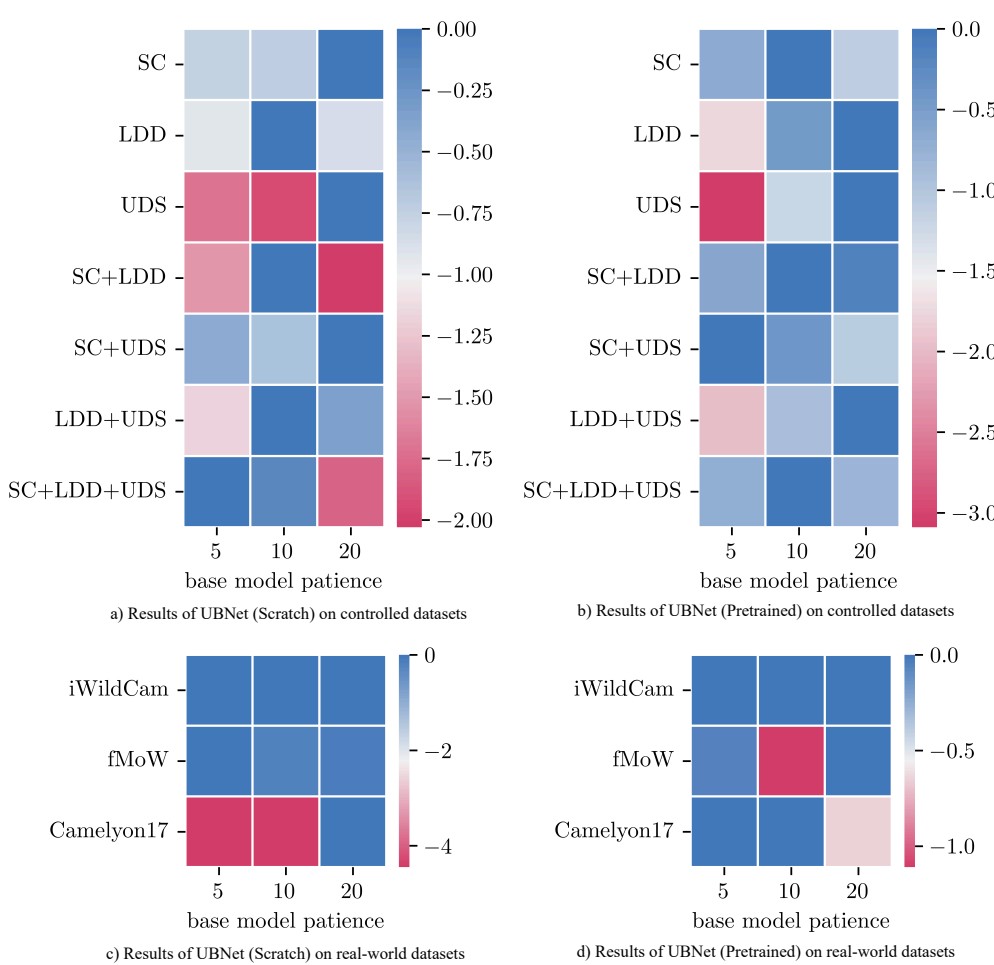

Figure 30: **Hyperparameter Exploration for UBNet**.

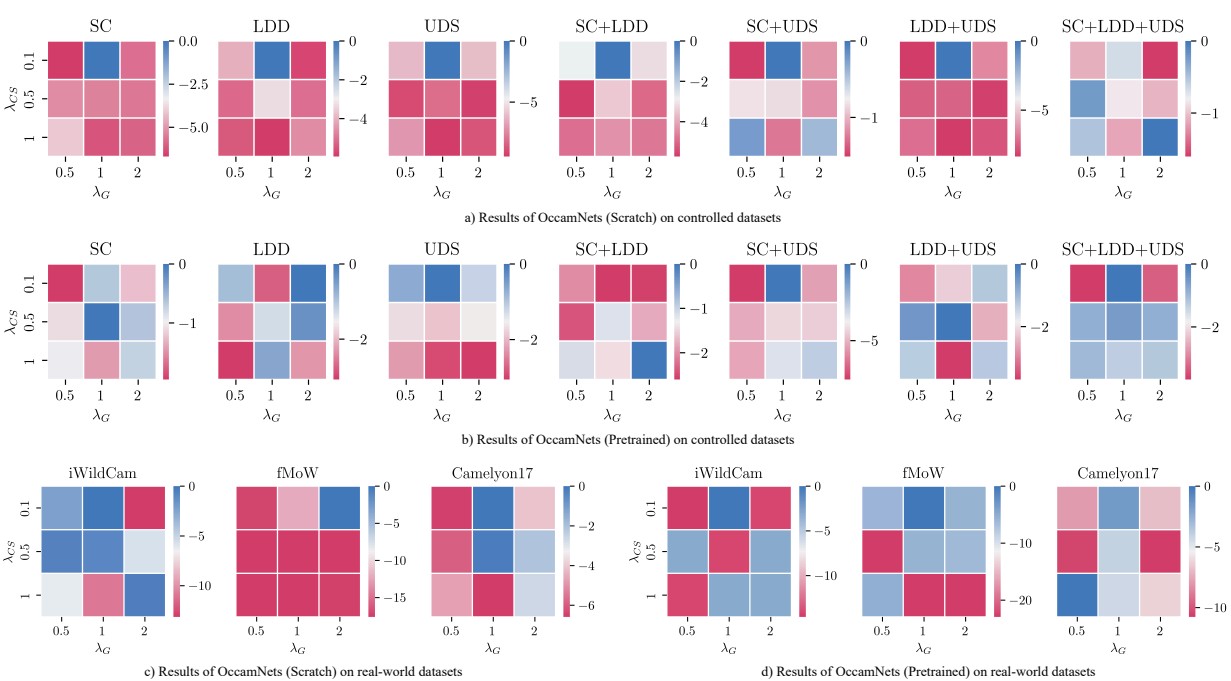

Figure 31: **Hyperparameter Exploration for OccamNets**.

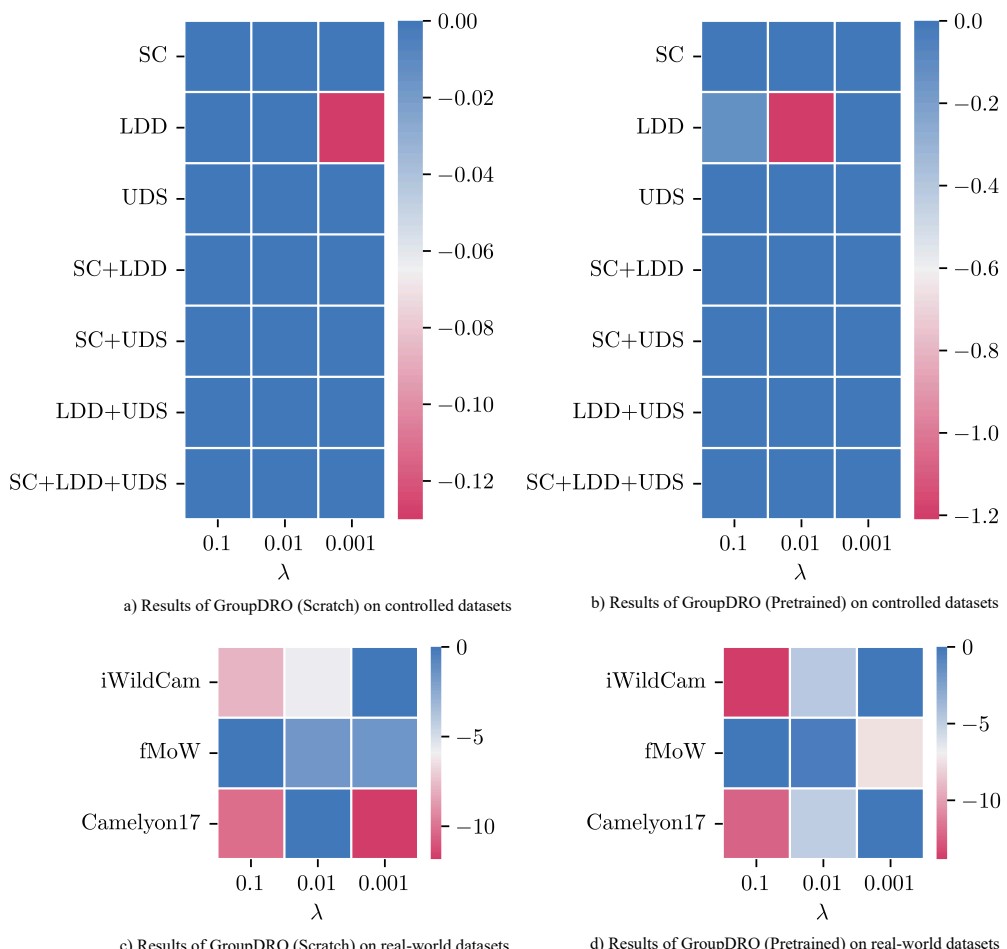

a) Results of GroupDRO (Scratch) on controlled datasets

b) Results of GroupDRO (Pretrained) on controlled datasets

c) Results of GroupDRO (Scratch) on real-world datasets

d) Results of GroupDRO (Pretrained) on real-world datasets

Figure 32: **Hyperparameter Exploration for groupDRO**.

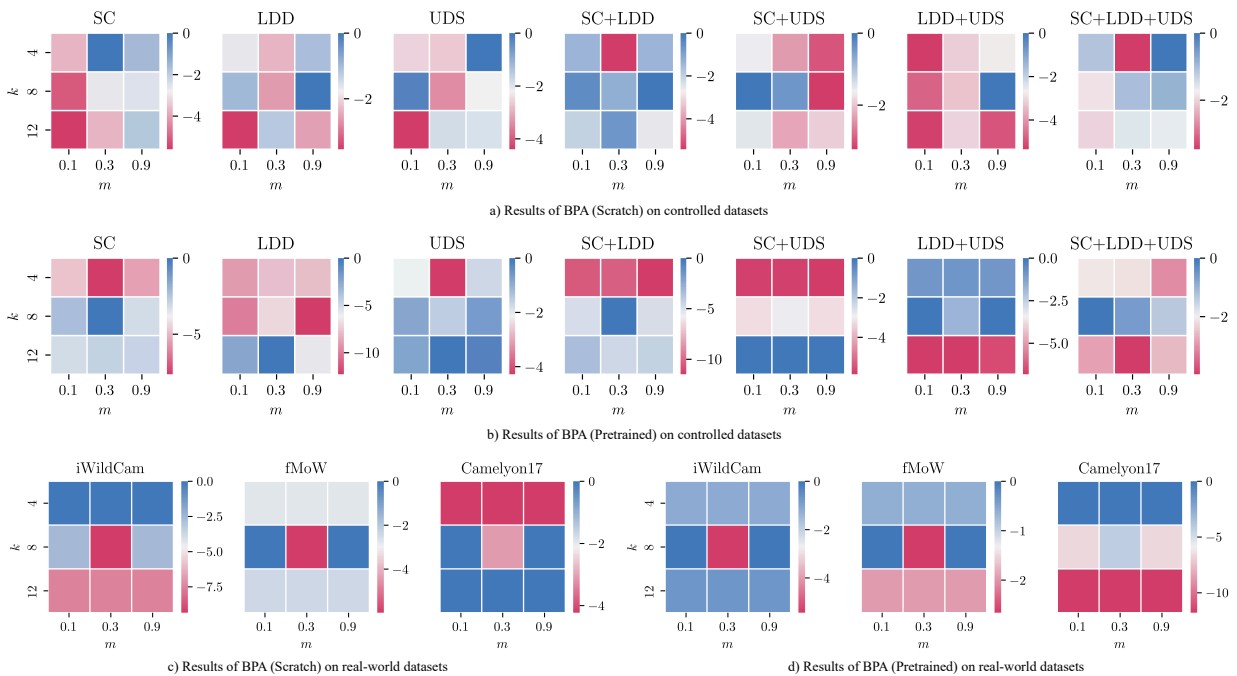

Figure 33: **Hyperparameter Exploration for BPA**.

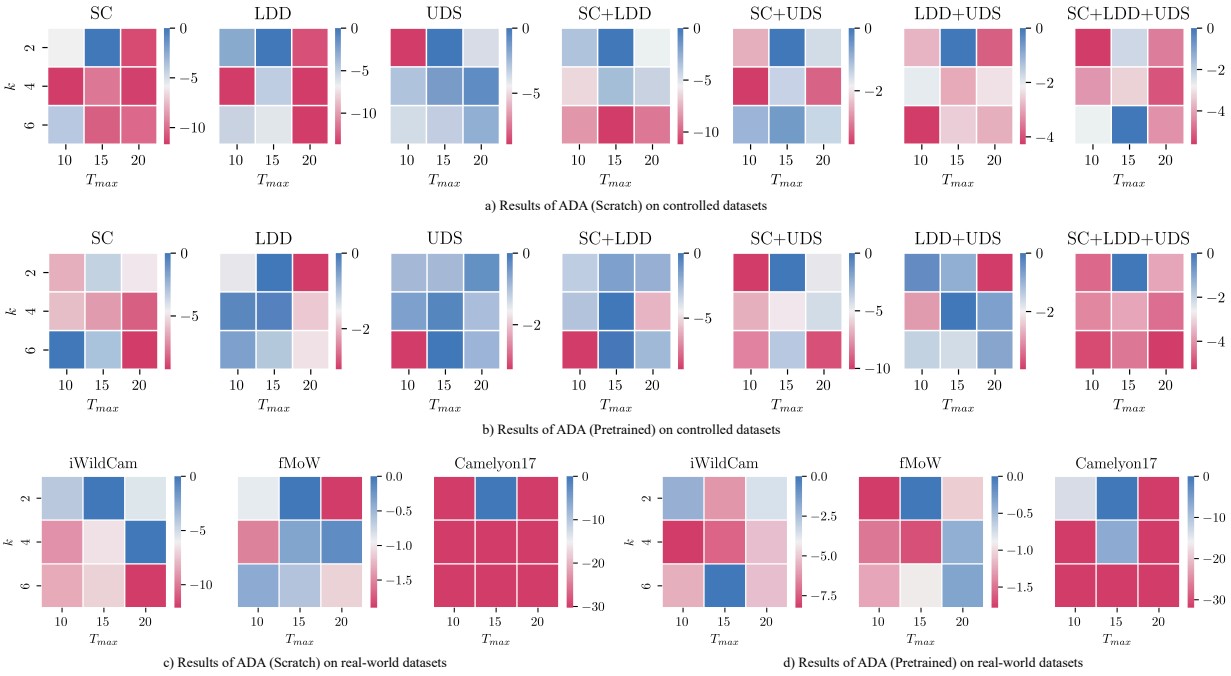

Figure 34: **Hyperparameter Exploration for ADA**.

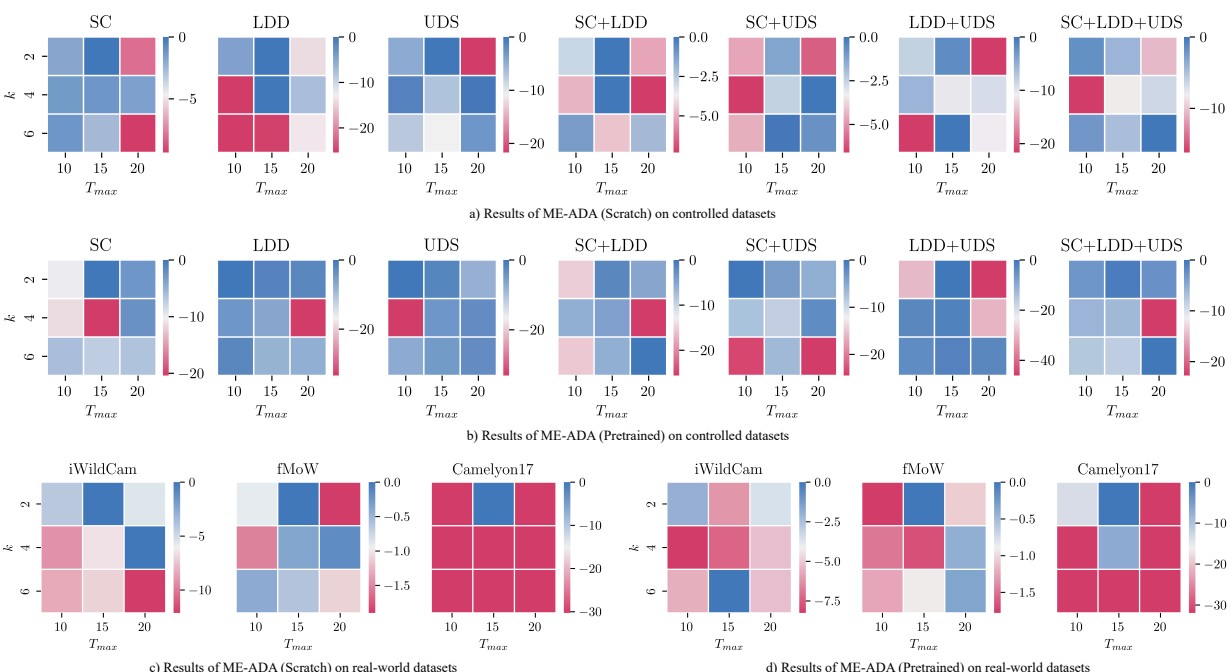

Figure 35: **Hyperparameter Exploration for ME-ADA**.

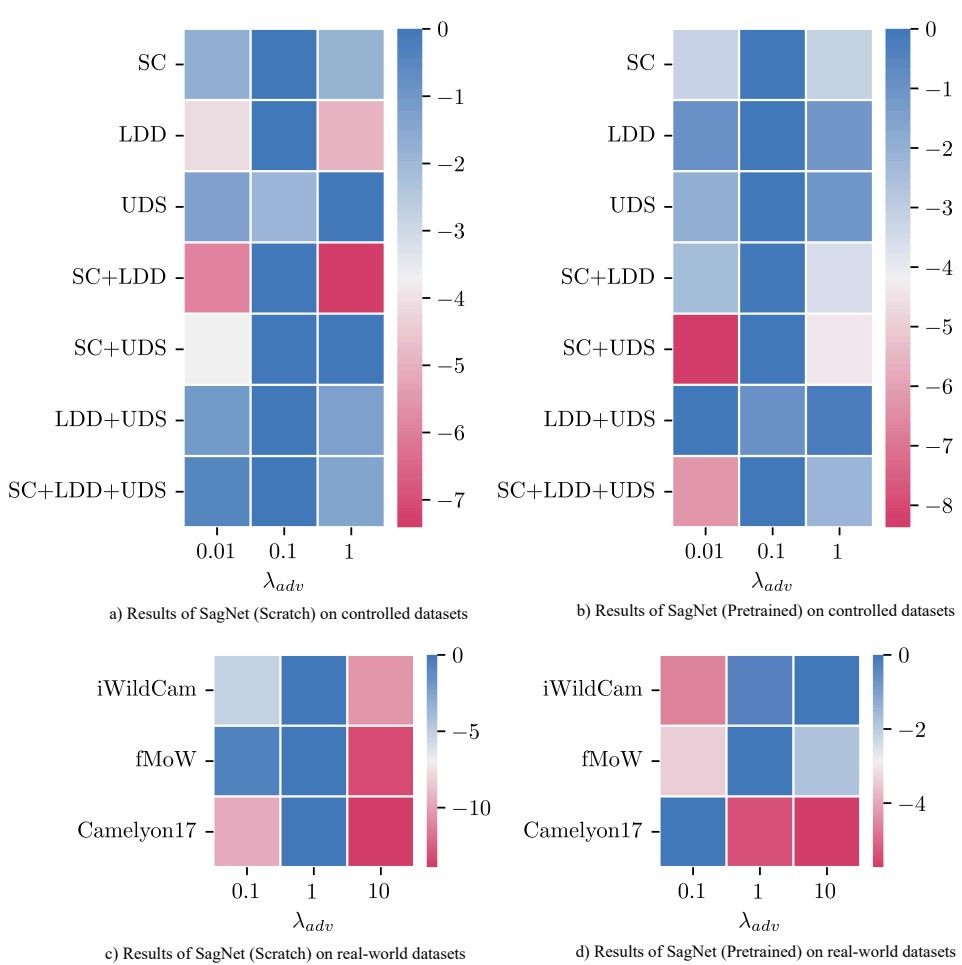

Figure 36: **Hyperparameter Exploration for SagNet**.

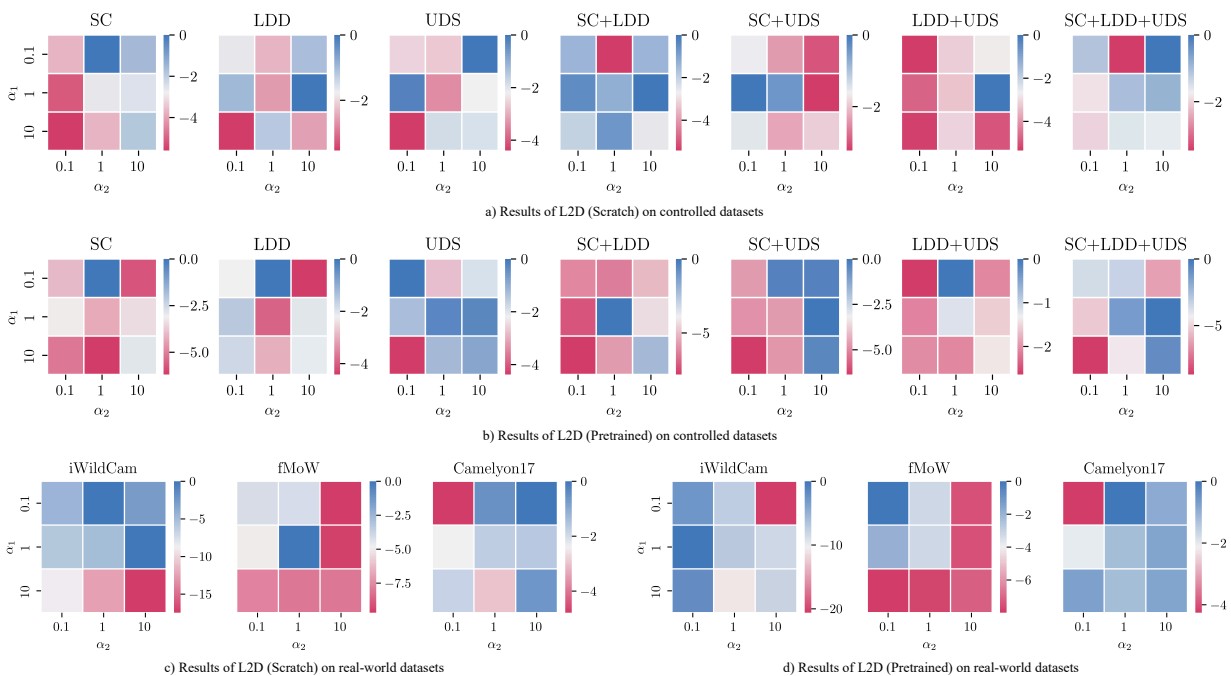

Figure 37: **Hyperparameter Exploration for L2D**.

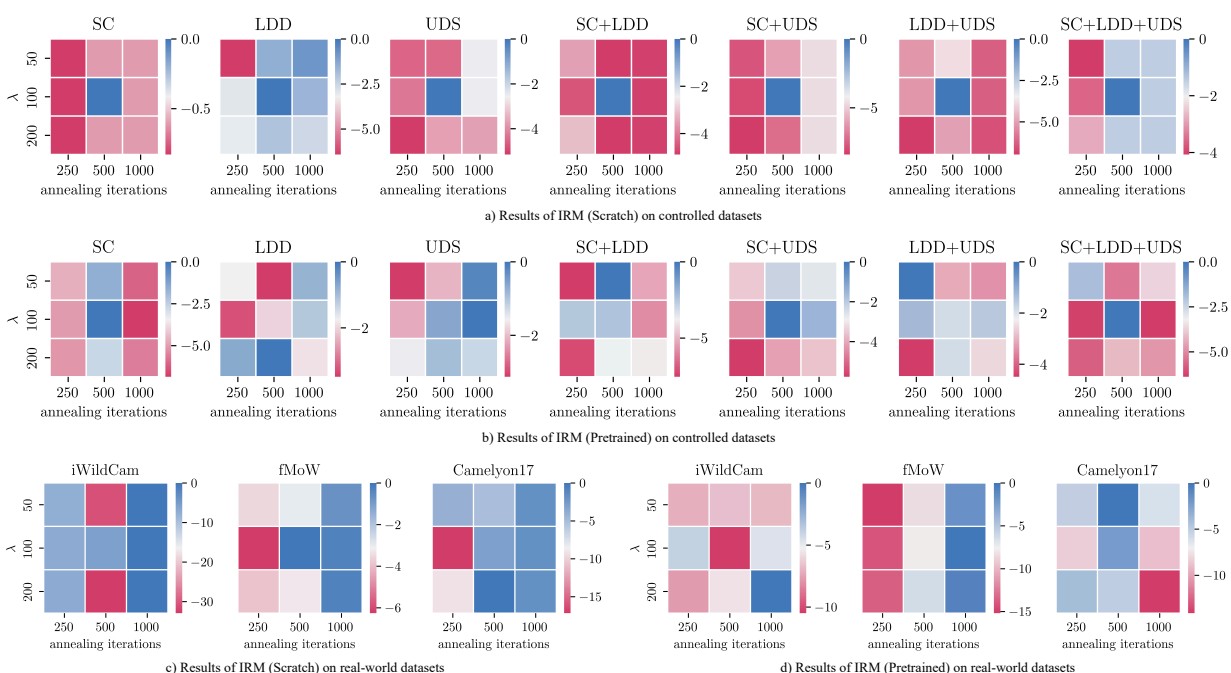

Figure 38: **Hyperparameter Exploration for IRM**.

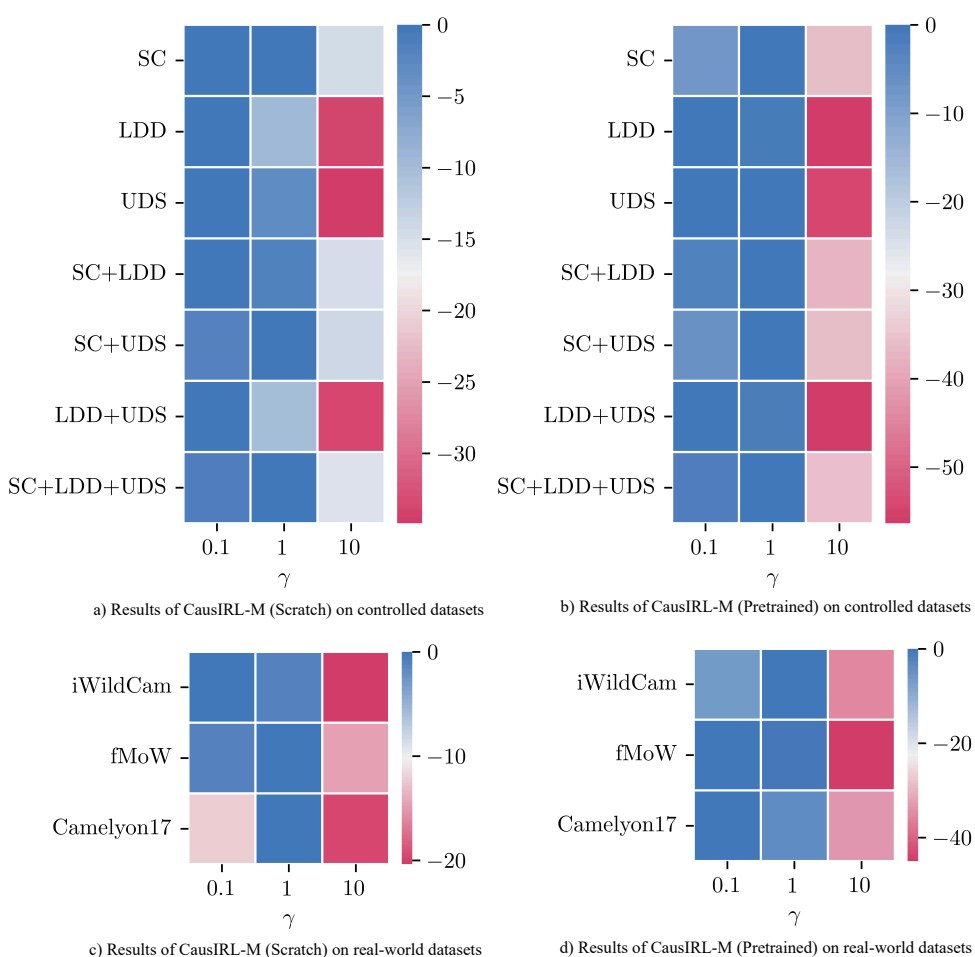

a) Results of CausIRL-M (Scratch) on controlled datasets

b) Results of CausIRL-M (Pretrained) on controlled datasets

c) Results of CausIRL-M (Scratch) on real-world datasets

d) Results of CausIRL-M (Pretrained) on real-world datasets

Figure 39: **Hyperparameter Exploration for CausIRL-M**.

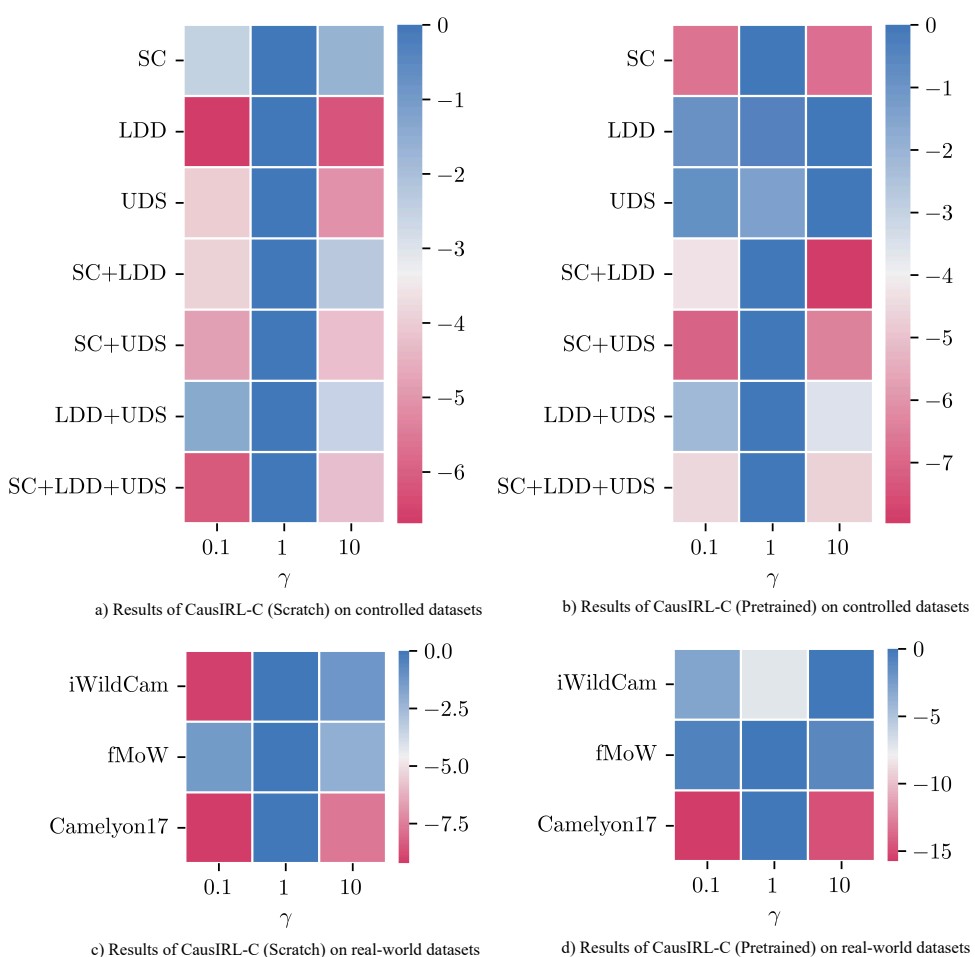

Figure 40: **Hyperparameter Exploration for CausIRL-C**.

### B.11 Computaional Cost

We provide the time and memory costs for all the algorithms, using iWildCam as the dataset for this analysis. Table 16 shows the results.

| Category | Method | Time (seconds per epoch) | Memory (MiB) |
|---|---|---|---|
| Architecture | ResNet18 | 1275 | 3274 |
| | ResNet50 | 1270 | 11640 |
| | ResNet101 | 1289 | 17179 |
| | ViT | 1258 | 17844 |
| Augmentation | ImageNet | 1405 | 11640 |
| | AugMix | 1617 | 11640 |
| | RandAug | 1418 | 11640 |
| | AutoAug | 1357 | 11640 |
| SDG | ADA | 1301 | 11487 |
| | ME-ADA | 1305 | 11487 |
| | SagNet | 1281 | 11718 |
| | L2D | 1607 | 24393 |
| OOD | IRM | 1272 | 11640 |
| | CausIRL-MMD | 1275 | 11563 |
| | CausIRL-CORAL | 1286 | 11606 |
| De-bias | UBNet | 1262 | 11489 |
| | PnD | 1658 | 62225 |
| | OccamNets | 3218 | 14669 |
| Worst-case | GroupDRO | 1258 | 11562 |
| | BPA | 1249 | 12235 |
| LVLM (LoRA) | InstructBLIP | 50 | 70567 |
| | LLaVA-1.5 | 308 | 110880* |
| | Phi-3.5-vision | 577 | 12623 |

Table 16: **Performance comparison of different methods across categories, showing time (in seconds) and memory usage (in MiB).** Training was performed with a batch size of 128 on a single H100 GPU. LVLM models are fine-tuned using LoRA for Table 14. For LLaVA-1.5, due to memory constraints on a single H100, multi-GPU training was used with two H100s, each handling a batch size of 64, and the total memory usage was combined.

