# OpenReview forum: "An Analysis of Model Robustness across Concurrent Distribution Shifts"
_TMLR — Accepted by TMLR_

### Review · Reviewer_yz3M · 2024-10-15

**Summary Of Contributions:**

This paper expands on the traditional concept of distribution shifts (DSs) by defining Concurrent Distribution Shifts (ConDS), where multiple types of shifts occur simultaneously within the dataset. This novel framework provides a more realistic representation of real-world data complexities compared to traditional Unique Distribution Shifts (UniDS), where only a single type of shift is analyzed.

**Audience:**

Yes

**Claims And Evidence:**

Yes

**Requested Changes:**

1. The inclusion of more real-world datasets with diverse attributes or using data augmentation techniques to create additional attributes would significantly enhance the framework’s relevance and applicability.
2.  Consider exploring the use of learned attributes or unsupervised feature extraction to induce distribution shifts. This could be implemented using models like autoencoders or self-supervised learning techniques.
3. Provide more in-depth analysis on why foundation models underperform in real-world scenarios compared to controlled datasets. This could include experiments with different training regimes or prompts that might mitigate these issues.
4. Include a section that explores how sensitive the results are to hyperparameter changes, particularly under complex distribution shifts like ConDS. This could involve reporting performance variations across different parameter settings for key algorithms.
5. Add clarifications on the benchmarking methodology, such as the choice of metrics used for evaluation and any considerations for dataset standardization. A table summarizing these choices and rationales could be helpful.

**Strengths And Weaknesses:**

1. The introduction of Concurrent Distribution Shifts (ConDS) is a novel contribution that reflects the complexities of real-world data more accurately than traditional single-shift frameworks. This approach provides a new perspective on evaluating machine learning models under more realistic conditions.
2. The inclusion of both synthetic datasets (e.g., dSprites) and real-world datasets (e.g., CelebA, DeepFashion) ensures that the framework is tested across diverse contexts. This enhances the generalizability and applicability of the findings to different domains.
3. The study’s findings are well-supported by detailed analysis, highlighting that heuristic augmentations and pre-training are particularly effective across shifts. This provides practical guidance for researchers and practitioners looking to enhance model robustness.

Weaknesses

1. While the paper covers a diverse set of datasets, the real-world datasets used have limited attributes for creating complex distribution shifts. This somewhat constrains the applicability of the findings to more varied real-world conditions. Future research could address this by incorporating additional real-world datasets or employing techniques to generate or simulate more attributes, thereby expanding the framework’s practical relevance.

2. The reliance on pre-annotated attributes to create distribution shifts may limit the framework's applicability to datasets without such detailed annotations. This approach may not capture more subtle or complex shifts that emerge from latent features or learned representations.The authors could consider exploring methods that incorporate learned attributes, which could make the framework more adaptable and applicable to a broader range of datasets.

3. The paper finds that foundation models, while effective in controlled settings, tend to underperform in real-world datasets. This is a significant limitation, as it highlights the gap between theoretical generalization and practical applicability in diverse domains. Additional strategies for tailoring foundation models to specific real-world contexts could enhance the robustness and practical utility of these models.

4. While the paper evaluates a wide range of algorithms, there is limited discussion on adapting specific algorithms to better handle ConDS. Addressing this aspect could offer more tailored insights and potentially improve performance under complex shifts.

5. The results might be sensitive to hyperparameter settings, especially in scenarios involving complex distribution shifts. An explicit analysis of hyperparameter sensitivity would provide greater transparency and make the results more robust.

---

> ### Author Response · Authors · 2024-11-23
>
> Thank you for your review and suggestions. Please find our responses below.
>
>
> ### Q1. The inclusion of more real-world datasets with diverse attributes or using data augmentation techniques to create additional attributes would significantly enhance the framework’s relevance and applicability.
>
> Our framework requires datasets with rich attribute annotations to create ConDS. However, such datasets are limited as annotations are expensive. We did consider using augmentations to create additional attributions, but augmentation techniques in algorithm baselines might directly address these shifts in this setting. However, for the rebuttal, we applied three types of corruptions from ImageNet-C [1]—impulse noise, snow, and elastic transform—on CelebA. Each attribute was evaluated under two conditions (corrupted and uncorrupted). The experimental results are presented in Section B.8 of the Appendix.
>
> [1] Hendrycks, Dan, et al. "Benchmarking Neural Network Robustness to Common Corruptions and Perturbations." ICLR 2018.
>
> ### Q2. Consider exploring the use of learned attributes or unsupervised feature extraction to induce distribution shifts.
>
> We propose leveraging the learned representation space and clustering techniques for this approach. Taking CelebA as an example, the procedure would involve the following steps:
>
> - Feature Encoding: Extract feature vectors using the Vision foundation model (e.g., Dino2 ViT).
> - Clustering: Apply K-Means clustering (K = 3) to each label category (e.g., “male” and “female”), resulting in clusters 1, 2, 3 for “male” and 4, 5, 6 for “female”.
> - Dataset Splitting: Define training and testing sets by creating combinations of clusters. For training, use combinations such as (1,4), (2,4), (1,5), and (2,5). The test set is defined using clusters (3,6).
> - Result Averaging: Compute the final results as the average performance across all splits.
>
> This setup assumes that instances grouped within the same cluster share similar features or attributes. Consequently, splitting the clusters into training and testing sets induces DSs, enabling an evaluation of generalization under these conditions. The results are detailed in Section B.9 of the Appendix.
>
> ### Q3. Provide more in-depth analysis on why foundation models underperform in real-world scenarios compared to controlled datasets.
>
> Our evaluation on real-world datasets like Camelyon17, which contains complex cell images, iWildCam with its camera trap images of diverse animal species, and FMoW's satellite images present unique challenges due to their niche content and visual complexity. This high level of domain specificity, with features likely outside the foundation models' general scope, limits their capacity to generalize effectively, particularly in zero-shot settings.
> An intuitive approach to evaluate this is by fine-tuning the vision encoder on these specialized datasets. Through fine-tuning, we found that the foundation models performed as expected, showing high effectiveness for these datasets. The results are presented in Table 14 in the Appendix.
>
> | Model            | iWildCam (w/o) | iWildCam (w) | Camelyon17 (w/o) | Camelyon17 (w) | FMoW (w/o) | FMoW (w) |
> |-------------------|----------------|--------------|------------------|----------------|------------|----------|
> | LLaVA-1.5        | 4.64           | **91.12**    | 51.09           | **95.32**     | 16.32  | **72.67**    |
> | Phi-3.5-Vision   | 9.91           | **91.19**    | 66.71           | **93.35**     | 10.88      | **77.02**|
> | InstructBLIP     | 1.86           | **12.13**    | 68.03           | **99.87**     | 18.17      | **41.18**|
>
> This result aligns with our finding that strong representations (achieved through ImageNet pre-training) are effective in enhancing robustness, as foundation models also benefit from extensive data pre-training.
>
> ### Q4. Include a section that explores how sensitive the results are to hyperparameter changes, particularly under complex distribution shifts like ConDS.
>
> The experiments with different parameter settings are presented in Section B.10 of the Appendix.
>
> ### Q5. Add clarifications on the benchmarking methodology, such as the choice of metrics used for evaluation and any considerations for dataset standardization.
>
> Thank you to the reviewer for this suggestion. We have added the benchmarking methodology in Section A.4 of the Appendix.

---

> > ### Comment · Reviewer_yz3M · 2024-12-17
> > **response to the author**
> >
> > Thank you for your detailed responses and additional experiments. I appreciate the effort you've made to address the concerns. The technical approach of your paper is sound, and the topic is quite interesting. I am inclined to recommend acceptance.re

---

### Review · Reviewer_JanM · 2024-10-28

**Summary Of Contributions:**

The paper presents an orthogonalized analysis on the impact of concurrent train/test data distribution shift. They focus their investigation on Spurious Correlation (SC), Low Data Drift (LDD) and Unseen Data Shift (UDS) and their impact on performance when multiple of these shifts appears between the training and testing set. They start from controllable (toy synthetic in some case) datasets where they have access to sample attributes and manually induce distribution shifts and study its impact on performance. They also include well known real-world distribution shift datasets (from WILDS) in their study. The authors then apply a wide array of robustness techniques and foundation models to these tasks. They observe that:
* Concurrent distribution shift is more challenging and isolated shift, although most of the challenge comes from spurious correlations
* If a methodology helps for one setting it tends to help for others as well
* Heuristic augmentations and pre-training (both imagenet pre-training and zero-shot foundation model levels) are effective
* Generalization algorithms provide limited performance gains

**Audience:**

Yes

**Broader Impact Concerns:**

I do not see concerns on ethical implications of this paper.

**Claims And Evidence:**

Yes

**Requested Changes:**

1. You report that the experiments were ran with multiple seeds and different attribute-shift pairings. Please report error bars in plots and tables.
2. Running experiments with some sort of adaptation of foundation model to the real-world datasets (See point 1 of Weakness).
3. Add clarification to Appendix section A.2. Specifically what is a split, because it is not defined in Section 3. (See point 2 of Weakness).
4. Measure relative regressions of results use to make Takeaway 2 (See point 5 of Weakness)
5. More thorough related works (See point 6 in Weakness)

**Nit**
* In Section 1, the sentence for finding (i) is difficult to parse.
* In Table 2, it would make it much clearer for the attribute name to be in the tables along with the values they can take.
* If possible, a very rough estimate of the total compute time would be great to have in there alongside the type of compute used.

**Strengths And Weaknesses:**

**Strengths**
* The paper reads very well, the story is easy to follow and results are presented clearly such that the conclusions are easy to observe as a reader
* Research approach is systematic and offers a spectrum of experimentation from very toy-like to real-world settings
* Conclusions that relate to concurrent distribution shifts are interesting — notably that if a methodology helps for one setting it tends to help with others.

**Weakness**
1. Foundation models for real-world dataset can be misleading since the tasks are so niche (e.g., cancer detection, satellite image). Perhaps a few-shot or fine-tuning stage would be required for this set of results.
2. Appendix section A.2 is slightly confusing. It seems the dataset sizes are pretty small (<100 samples in some cases?). Could you comment on how the dataset scale (assuming the same shift) might impact your results?
3. Takeaway 1: “ConDS is, on average, more challenging than UniDS”, could’ve be explored in a bit more depth. This setting with concurrent shifts being more challenging is somewhat expected, but finding out how these shifts interact together would be very useful for the community — are they additive, compounding? Could you comment a bit on that here?
4. The authors does not discuss possible higher level shift that might occur when inducing shifts across different attributes. Is it possible that setups with concurrent shifts across different attributes induces additional shifts of attribute pairs that could significantly influence results? That possibility should be discussed in the results. Could you comment on this here?
5. In Takeaway 2: “We see that the presence of SC tends to dominate over other DSs in ConD” was only looked through the lens of absolute performance regressions. The correct measure here would be comparing relative regressions (UNIFORM -> LSS+UDS VS SC -> SC+LDD+UDS).
6. Related works could use a bit more work, specifically around literature that investigates the types of distribution shifts, e.g., [1], [2], [3]

[1]Ye, Nanyang, et al. "Ood-bench: Quantifying and understanding two dimensions of out-of-distribution generalization." Proceedings of the IEEE/CVF Conference on Computer Vision and Pattern Recognition. 2022.

[2] Yao, Huaxiu, et al. "Wild-time: A benchmark of in-the-wild distribution shift over time." Advances in Neural Information Processing Systems 35 (2022): 10309-10324.

[3] Recht, Benjamin, et al. "Do imagenet classifiers generalize to imagenet?." International conference on machine learning. PMLR, 2019.

---

> ### Author Response · Authors · 2024-11-23
>
> Thank you for your review and suggestions. Please find our responses below.
>
>
> ### Q1. You report that the experiments were ran with multiple seeds and different attribute-shift pairings. Please report error bars in plots and tables.
>
> We reported error bars in Table 3, 11, and Figure 4 in the updated script.
>
> ### Q2. Running experiments with some sort of adaptation of foundation model to the real-world datasets.
>
> Thank you for this suggestion. Through fine-tuning, we found that the foundation models performed as expected, showing high effectiveness for these datasets. The results are presented in Table 14 in the Appendix B.6.
>
> | Model            | iWildCam (w/o) | iWildCam (w) | Camelyon17 (w/o) | Camelyon17 (w) | FMoW (w/o) | FMoW (w) |
> |-------------------|----------------|--------------|------------------|----------------|------------|----------|
> | LLaVA-1.5        | 4.64           | **91.12**    | 51.09           | **95.32**     | 16.32  | **72.67**    |
> | Phi-3.5-Vision   | 9.91           | **91.19**    | 66.71           | **93.35**     | 10.88      | **77.02**|
> | InstructBLIP     | 1.86           | **12.13**    | 68.03           | **99.87**     | 18.17      | **41.18**|
>
> This result aligns with our finding that strong representations (achieved through ImageNet pre-training) are effective in enhancing robustness, as foundation models also benefit from extensive data pre-training.
>
> ### Q3. Add clarification to Appendix section A.2. Specifically what is a split, because it is not defined in Section 3.
>
> We outlined our approach for creating DSs using the five controlled datasets. Additionally, we commit to releasing our data splits when this submission is deanonymized. We added the description in Section A.2 of the Appendix.
>
> ### Q4. Measure relative regressions of results use to make Takeaway 2
>
> Thank you for this suggestion. We compared (i)(Uniform, LDD) with(SC, SC+LDD), (ii)(Uniform, UDS) with(SC, SC+UDS), and (iii)(Uniform, LDD+UDS) with(SC, SC+LDD+UDS). These observations support our argument: although ConDS introduces more challenges than UniDS for both LDD and UDS, there is a trivial accuracy drop when transitioning from SC to SC+LDD or SC+UDS.
>
> | Distribution Shisft          | Performance Gap | Distribution Shisft          | Performance Gap |
> |-----------------------|--------------|----------------------|--------------|
> | (Uniform, LDD)      | 1.05%        | (SC, SC+LDD)       | 0.32%        |
> | (Uniform, UDS)      | 2.09%        | (SC, SC+UDS)       | 0.15%        |
> | (Uniform, LDD+UDS)  | 3.89%        | (SC, SC+LDD+UDS)   | 1.06%        |
>
> ### Q5. More thorough related works
>
> Thank you for pointing out these related works. We have reviewed and expanded on the suggested papers and incorporated them into related work (see Section 2).

---

> ### Author Response · Authors · 2024-11-23
>
> ### W2. It seems the dataset sizes are pretty small (<100 samples in some cases?). Could you comment on how the dataset scale (assuming the same shift) might impact your results?
>
> We could not use many samples because there are insufficient image samples to make DSs in some datasets. Instead, we additionally conduct experiments with various dataset sizes for Shapes3D, and CelebA. The results for these three dataset size variations are provided in Section B.3 the Appendix. Although the overall trend remains similar, some differences are observed.
>
> ### W3. This setting with concurrent shifts being more challenging is somewhat expected, but finding out how these shifts interact together would be very useful for the community — are they additive, compounding?
>
> Using the results in Figure 4, we calculated the performance gap (PG) for each DS relative to the Uniform (i.i.d.) distribution. A comparison of PG(Uniform, LDD) and PG(Uniform, UDS) with PG(Uniform, LDD+UDS) shows that concurrent shifts increase the difficulty of the task. However, when LDD and UDS are combined with SC, the impact becomes negligible. Given that the PG for SC is substantially larger than for either LDD or UDS, we conclude that SC dominates the influence of other shifts.
>
> | Distribution Shisft          | Performance Gap | Distribution Shisft          | Performance Gap | Distribution Shisft          | Performance Gap |
> |-----------------------|--------------|----------------------|--------------|----------------|--------------|
> | (Uniform, SC)      | 14.78%        | (Uniform, SC+LDD)       | 15.10%    | (Uniform, SC+LDD+UDS)    | 15.84%
> | (Uniform, LDD)      |  1.05%       | (Uniform, SC+UDS)       | 14.93%      | - | -  |
> | (Uniform,UDS)  |  2.09%      | (Uniform, LDD+UDS)   | 3.89%       |  - | - |
>
>
> ### W4. Is it possible that setups with concurrent shifts across different attributes induces additional shifts of attribute pairs that could significantly influence results?
>
> Thank you for raising this important point. The concurrent shifts across different attributes can indeed induce additional shifts. Consider the setup and attributes shown in Figure 2. Take for example the SC+LDD shift, created by two attributes e.g., object color and background color (see Fig. 1-right, for examples). It is possible that there are other attributes e.g., object size, is correlated to the object’s color. This indeed results in additional distribution shifts and potentially worse performance.
>
> However, we do not foresee this occurring for our evaluations on controlled datasets like dSprites, Shapes3D, as the attributes are known and we sample uniformly from attributes that are not used to induce distribution shifts. For real-world datasets like FMoW, we do not have rich attribute annotations or knowledge about the data generation process; it is possible that there are additional shifts beyond the ones indicated in the dataset (UDS+LDD). Thus, our ablation studies that require control over different attributes e.g., performance under increasing number of shifts (Fig. 4), are only done on controlled datasets.
>
> ### Nit
>
> We have revised *the sentence for finding (i)* and *Table 2*, and the changes have been reflected in the uploaded script. The *experiments on compute time* are presented in Section B.11 of the Appendix.

---

### Review · Reviewer_6Dtq · 2024-11-09

**Summary Of Contributions:**

The paper introduces an evaluation framework for understanding the robustness of machine learning models under more than one real-world distribution shift attributes (concurrent domain shifts) because in real-world scenarios, models face multiple unseen domain shifts combined with spurious correlations.

### Contributions

1. The proposed work performs extensive ablation studies to investigate how various data distribution attributes contribute to concurrent domain shifts. To be specific, the authors evaluate 26 different algorithms across 168 source-target pairs containing both synthetic and real-world datasets.

2. The proposed study presents two interesting and important conclusions: first, concurrent domain shifts generally lead to poorer performance. However, models that improve generalization for one type of shift often perform well with other shifts as well. Second, heuristic data augmentations are more effective than many advanced generalization methods.

**Audience:**

Yes

**Claims And Evidence:**

Yes

**Requested Changes:**

1. Conducting more experiments and analyses on more complex and large-scale datasets, such as ImageNet and VisDA-2017, would help capture a wider range of uncontrollable distribution shifts.

2. Conducting more experiments using more recent large language models, such as GPT and Llama, for zero-shot evaluations.

3. Including a more detailed discussion on the limitations and potential failure modes of the evaluated algorithms, particularly regarding the heuristic data augmentation methods and their effectiveness in different scenarios.

4. Including specific case studies or examples that show how the proposed evaluation framework can be applied to real-world problems with uncontrollable domain shifts.

5. Providing visualizations to better illustrate the experimental results. For example, t-SNE plots could show how domain shifts affect the clustering of latent features.

**Strengths And Weaknesses:**

### Strengths

1. The proposed ConDS evaluation framework offers a more realistic method to evaluating model robustness against the distribution shifts. By considering multiple domain shift attributes, the paper provides important insights that help the machine learning community better understand the domain generalization problems and how to handle them more effectively.

2. The evaluation includes a broad set of 26 algorithms and 168 source-target pairs across multiple datasets, offering a solid analysis of various types of domain shifts and how they affect the model generalization in general.

3. The paper provides a detailed analysis of the limitations of vision-language foundation models (e.g., CLIP and InstructBLIP) when applied to complex distribution shifts, contributing practical values to the community.

4. The paper is well-written with clear motivations.

### Weaknesses

1. While the paper covers a wide range of datasets, the study could benefit from exploring a wider range of uncontrolled domain shifts existing in more complex real-world datasets, like ImageNet or VisDA-2017.

2. The claim that heuristic augmentation methods are more effective than the advanced generalization techniques is in fact very strong, challenging many areas of machine learning. To convince practitioners, it is important to conduct experiments on more complex real-world datasets.

---

> ### Author Response · Authors · 2024-11-23
>
> Thank you for your review and suggestions. Please find our responses below.
>
>
> ### Q1. Conducting more experiments and analyses on more complex and large-scale datasets, such as ImageNet and VisDA-2017, would help capture a wider range of uncontrollable distribution shifts.
>
> Please note that our primary contribution in this submission is focused on understanding algorithms within the ConDS framework. While analysis with ImageNet and VisDA-2017 would indeed be insightful, these datasets reflect UniDS, and in particular, UDS, which has been extensively studied [1, 2, 3] and differs from our intended setup.
>
> [1] Hendrycks, Dan, et al. "Benchmarking Neural Network Robustness to Common Corruptions and Perturbations." ICLR 2018.
> [2] French, Geoff, et al. "Self-ensembling for visual domain adaptation." ICLR 2018.
> [3] Kang, Guoliang, et al. "Contrastive adaptation network for unsupervised domain adaptation." CVPR 2019.
>
> ### W2. The claim that heuristic augmentation methods are more effective than the advanced generalization techniques is in fact very strong, challenging many areas of machine learning. To convince practitioners, it is important to conduct experiments on more complex real-world datasets.
>
> We acknowledge that the observation of heuristic augmentation methods being more effective than certain generalization techniques is unexpected. However, our work extends the findings of [4], specifically that ‘pretraining and augmentations (whether learned or heuristic) yield substantial benefits in many cases,’ to the ConDS setting. We believe this insight will be valuable for practitioners.
>
> [4] Olivia Wiles, et al. "A fine-grained analysis on distribution shift." ICLR 2022.
>
>
> ### Q2. Conducting more experiments using more recent large language models, such as GPT and Llama, for zero-shot evaluations.
>
> Thank you for this suggestion. Consenting with Reviewer 6Dtq, we add *4o-mini* and *4o* in our baselines. The results are presented in all the result tables and figures including Figure 3 and Table 3.
>
> | Dataset     | GPT-4o-mini | GPT-4o   |
> |-------------|-------------|----------|
> | iWildCam    | 43.43      | 51.80   |
> | Camelyon17  | 64.98      | 58.10   |
> | FMoW        | 19.89      | 25.58   |
>
>
> ### Q3. Including a more detailed discussion on the limitations and potential failure modes of the evaluated algorithms, particularly regarding the heuristic data augmentation methods and their effectiveness in different scenarios.
>
> As presented in the script, while the methods perform well in their benchmark settings, they sometimes exhibit limitations in overall performance. Zero-shot inference with a foundation model is also limited when predicting unfamiliar data. Notably, learning-free and simple augmentation techniques prove effective. However, although heuristic data augmentation generally enhances robustness, it is not without flaws, as its effectiveness depends on how well it aligns with the attributes driving DSs.
>
> ### Q4. Including specific case studies or examples that show how the proposed evaluation framework can be applied to real-world problems with uncontrollable domain shifts.
>
> Real-world problems often exhibit ConDS. For ConDS, as highlighted in the script’s takeaways, we observed that pretraining and heuristic augmentations are generally more effective than generalization algorithms (see Section 5.3.1 Takeaway 4). Additionally, when using foundation models, prompts should be carefully tailored to the specific context (see Tables 4 and 5).
>
> ### Q5. Providing visualizations to better illustrate the experimental results. For example, t-SNE plots could show how domain shifts affect the clustering of latent features.
>
> Thank you for the suggestion. To address this, we visualized the latent features of all algorithms using UMAP (chosen over T-SNE due to its lower computational cost). The visualization was performed for models trained on SC + LDD + UDS to examine whether the algorithms exhibit attribute invariance. We compared the latent feature distributions of the best- and worst-performing models on CelebA. We additionally explored learning from scratch vs. pre-training. The figures are presented in Section B.7 of the Appendix. While all the algorithms demonstrate invariance to LDD and UDS, ViT exhibits sensitivity to SC. In contrast, both CLIP-Large and ImageNet remain invariant to all DSs. In Figure 22, the ViT model with pre-training exhibits better invariance to attributes compared to the ViT model trained from scratch.

---

> > ### Comment · Reviewer_6Dtq · 2024-12-08
> > **Post rebuttal**
> >
> > The authors have addressed all of my concerns. I would like to recommend acceptance.

---

### Author Response · Authors · 2024-11-23

We sincerely thank all three reviewers for their valuable comments and feedback. We have addressed the reviewers' concerns as detailed below. Revisions have been incorporated in the manuscript and the changes made are highlighted in $\color{orange}{\text{orange}}$ (Reviewer 6Dtq), $\color{blue}{\text{blue}}$ (Reviewer JanM), and $\color{red}{\text{red}}$ (Reviewer yz3M). In particular, we would appreciate it if AC and the reviewers could note that many new experiments have been added to the Appendix.

---

### Decision · Action_Editor_CWUu · 2024-12-24

**Recommendation:** Accept as is

**Comment:**

This paper performs an extensive study of robustness of various models across concurrent distribution shifts. Contrary to prior work that focused on evaluating robustness to one distribution shift at a time, this work explores the behavior of various models against multiple concurrent shifts. They conduct this analysis first in a controlled (synthetic) environment and then in more realistic datasets (commonly-used datasets from WILDS), and draw various conclusions. Notably, they find that indeed, concurrent shifts are for the most part more challenging than single shifts. They also find that robustness against one type of shift transfers to other types and that heuristic augmentations perform strongly.

The reviewers found that the paper "reads very well [...] and results are presented clearly such that the conclusions are easy to observe as a reader" (Reviewer JanM).

They also found that the proposed framework "offers a more realistic method to evaluating model robustness" (Reviewer 6Dtq) and "provides a new perspective on evaluating machine learning models under more realistic conditions" (Reviewer yz3M). They also found that the paper "provides important insights" to better understand domain generalization issues and how to handle them, "contributing practical values to the community" (Reviewer 6Dtq).

The reviewer also noted that the analysis is comprehensive ("a solid analysis of various types of domain shifts", Reviewer 6Dtq; "a detailed analysis of the limitations of vision-language foundation models", Reviewer 6Dtq; "tested across diverse contexts" (Reviewer yz3M)) and that "the research approach is systematic and offers a spectrum of experimentation from very toy-like to real-world settings" (Reviewer JanM). Reviewer yz3M noted that "the study’s findings are well-supported by detailed analysis".

The reviewers also identified some weaknesses, many of which were addressed through the rebuttal. Notably, the reviewer thought the paper could benefit from more complex real-world datasets (Reviewer 6Dtq, Reviewer yz3M), more recent large models (Reviewer 6Dtq), the inclusion of more experiments that adapt foundation models to the niche datasets considered before evaluation (Reviewer JanM). The reviewers were satisfied with the discussion during the rebuttal and the authors conducted additional experiments too e.g. to demonstrate the improved performance of foundation models on niche tasks after an adaptation phase. Other concerns that were addressed during the rebuttal include additional discussion of limitations, of the size of datasets, inclusion of additional details on hyperparameters and dataset setup, additional visualizations, and various clarifications.

All reviewers either recommend acceptance (or learn towards acceptance). The remaining weakness noted in the recommendation of Reviewer 6Dtq is the missing experiments on large-scale datasets. Overall, this paper meets the bar of acceptance for TMLR based on the above discussion.

**Audience:**

This paper is on the topic of robustness to (concurrent) distribution shifts, which is of interest to TMLR's community. The reviewers noted the usefulness of this work in terms of providing a new perspective of evaluating robustness under more realistic conditions, and that the findings will constitute practical guidance for researchers and practitioners.

**Claims And Evidence:**

This paper performs an extensive study of robustness of various models across concurrent distribution shifts. Contrary to prior work that focused on evaluating robustness to one distribution shift at a time, this work explores the behavior of various models against multiple concurrent shifts. They conduct this analysis first in a controlled (synthetic) environment and then in more realistic datasets (commonly-used datasets from WILDS), and draw various conclusions. Notably, they find that indeed, concurrent shifts are for the most part more challenging than single shifts. They also find that robustness against one type of shift transfers to other types and that heuristic augmentations perform strongly.

The reviewers did find that the claims are supported by sufficient evidnce. Specifically, they found that the paper contains "extensive ablation studies", "a broad set of 26 algorithms and 168 source-target pairs across multiple datasets, offering a solid analysis" and "a detailed analysis of the limitations of vision-language foundation models" (Reviewer 6Dtq). Reviewer JanM also noted that the "research approach is systematic and offers a spectrum of experimentation from very toy-like to real-world settings" and Reviewer yz3M noted that "[...] the framework is tested across diverse contexts. This enhances the generalizability and applicability of the findings to different domains." and that "The study’s findings are well-supported by detailed analysis".